# Caspase-1 in *Cx3cr1*-expressing cells drives an IL-18-dependent T cell response that promotes parasite control during acute *Toxoplasma gondii* infection

**Isaac W. Babcock, Lydia A. Sibley, Sydney A. Labuzan, Maureen N. Cowan, Ish Sethi, Seblework Alemu, Abigail G. Kelly, Michael A. Kovacs, John R. Lukens, Tajie H. Harris** ⓘ *

Center for Brain Immunology and Glia, Department of Neuroscience, University of Virginia, Charlottesville, Virginia, United States of America

* tajieharris@virginia.edu

**Data Availability Statement:** The authors confirm that all data underlying the findings are fully available without restriction. All relevant data are

## Abstract

Inflammasome activation is a robust innate immune mechanism that promotes inflammatory responses through the release of alarmins and leaderless cytokines, including IL-1α, IL-1β, and IL-18. Various stimuli, including infectious agents and cellular stress, cause inflammasomes to assemble and activate caspase-1. Then, caspase-1 cleaves targets that lead to pore formation and leaderless cytokine activation and release. *Toxoplasma gondii* has been shown to promote inflammasome formation, but the cell types utilizing caspase-1 and the downstream effects on immunological outcomes during acute *in vivo* infection have not been explored. Here, using knockout mice, we examine the role of caspase-1 responses during acute *T. gondii* infection globally and in *Cx3cr1*-positive populations. We provide *in vivo* evidence that caspase-1 expression is critical for, IL-18 release, optimal interferon-γ (IFN-γ) production, monocyte and neutrophil recruitment to the site of infection, and parasite control. Specifically, we find that caspase-1 expression in *Cx3cr1*-positive cells drives IL-18 release, which potentiates CD4+ T cell IFN-γ production and parasite control. Notably, our *Cx3cr1-Casp1* knockouts exhibited a selective T cell defect, mirroring the phenotype observed in *Il18* knockouts. In further support of this finding, treatment of *Cx3cr1-Casp1* knockout mice with recombinant IL-18 restored CD4+ T cell IFN-γ responses and parasite control. Additionally, we show that neutrophil recruitment is dependent on IL-1 receptor accessory protein (IL-1RAP) signaling but is dispensable for parasite control. Overall, these experiments highlight the multifaceted role of caspase-1 in multiple cell populations contributing to specific pathways that collectively contribute to caspase-1 dependent immunity to *T. gondii*.

## Author summary

When a cell undergoes inflammatory cell death, termed pyroptosis, cellular content is released and has the potential to stimulate immune responses. Our work highlights that in

within the paper and its Supporting information files.

**Funding:** This work was funded by National Institute of Health (grants R01NS112516 and R21NS128551 to THH, T32AI007496 to IWB, 5T32GM136615 to LAS, 5T32NS1156573 to SAL, and S10RR031633-01 to UVA Flow Cytometry Core) and a Pinn Scholars Award from the University of Virginia to THH. The funders had no role in study design, data collection and analysis, decision to publish, or preparation of the manuscript.

**Competing interests:** The authors have declared that no competing interests exist.

the context of *T. gondii* infection, distinct cell populations undergo pyroptosis each of which has different impacts on how the immune system responds. These findings suggest a collaborative effort of multiple cell types undergoing pyroptosis for optimal immunity to infection. Using a cell-type specific knockout to render *Cx3cr1*[+] cells incapable of undergoing pyroptosis, we find that pyroptosis in *Cx3cr1*[+] cells reinforces adaptive immune cell function, while other population's pyroptosis stimulates the recruitment of innate immune cells into the infected tissue. We go on to identify a specific molecule, IL-18, is released from *Cx3cr1*-expressing cells pyroptosis and augments a specific adaptive immune response. By reintroducing IL-18 into mice lacking Casp1 in *Cx3cr1*-expressing cells, we successfully restored adaptive immune cell function thereby facilitating the recovery of parasite control. This study outlines the impact of pyroptosis on immunity to *T. gondii* and stratifies the effects from separate cell populations and their associated downstream pathways.

## Introduction

Innate pathogen sensing via pattern recognition receptors (PRRs) instructs and mobilizes the immune system in response to infections. The diversity of PRRs and the molecules they recognize tailor the immune response generated. These PRRs include toll-like receptors (TLR), Nod-like receptors (NLR), and C-type lectin receptors [1]. Specifically, activation of some NLRs triggers the formation of a multi-protein complex known as the inflammasome. The effector protein of the inflammasome is caspase-1, which cleaves substrates including those that create pores in the cell membrane. Consequently, this process prompts inflammatory cell death and the release of immune-activating cytosolic contents, including leaderless cytokines which do not leave the cell through conventional secretion pathways and rely on proteolytic cleavage for activation [2,3]. Currently, it remains unclear how inflammasome activation is orchestrated across cell types during an infection and whether each cell type has the same ability to activate the inflammasome and release inflammatory cargo. To investigate this question, we employed an *in vivo* mouse model of *Toxoplasma gondii* infection, in which the activation of the inflammasome has been reported to result in the release of various immunostimulatory molecules.

*T. gondii* is an obligate intracellular parasite that can infect nearly all nucleated cells. The parasite replicates throughout the body of the host during the acute phase of infection and then resides in the brain for the duration of the host's life as a chronic infection. Host survival depends on the adaptive immune system's ability to produce interferon gamma (IFN-γ) [4–7]. IFN-γ signaling stimulates the expression and production of proteins that facilitate intracellular pathogen-killing mechanisms, enabling the control of the parasite [8–10]. The dependence on IFN-γ underscores the vital role of innate immune sensing, which initiates production of cytokines like IL-12 and IL-18 that bridge the innate the adaptive immune system. In mice, TLR11/12 molecules on dendritic cells recognize *T. gondii* profilin resulting in MyD88-dependent IL-12 release [11–14]. IL-12 acts on natural killer (NK), ILC1, and T lymphocyte cells to promote an IFN-γ response [15–18]. Of note, humans lack functional TLR11/12 and can still mount an effective immune response to *T. gondii*, suggesting that other innate sensors recognize and respond to the parasite. One such sensor is the inflammasome, which has been suggested to promote protective immunity during acute infection in humans, rats, and mice [19–23]. At present, *in vitro* studies and survival studies have been performed but which cells utilize

caspase-1 at various stages of *T. gondii* infection and the immunological consequences have not been defined.

The inflammasome mediates the release of cytoplasmic cytokines (leaderless cytokines) and premade immunostimulatory molecules (alarmins). *In vitro* and *in vivo* studies have shown that *T. gondii* infection promotes the release of leaderless cytokines IL-1α, IL-1β, IL-33, and IL-18 and the alarmin s100a11 [19,20,22,24–29]. A recent study by López-Yglesias, et. al demonstrated that in the absence of TLR11, inflammasome effector proteins caspase-1/11 promote CD4[+] T cell IFN-γ production [21]. Another study observed that enhancing IL-18's ability to bind cell surface receptors can promote stronger CD4[+] T cell and NK cell IFN-γ production during *T. gondii* infection [30]. Additionally, it was reported that during acute infection IL-1 signaling onto its cognate receptor IL-1R is required for neutrophil recruitment to the site of infection, but this signal was dispensable for IFN-γ responses [25]. Furthermore, during acute infection, the alarmin s100a11 promotes a *Ccl2*-mediated monocyte recruitment to the site of infection. Together, these studies indicate that caspase-1/11 may play a role in multiple facets of the immune response to *T. gondii*. However, it remains uncertain whether caspase-1 specifically is necessary and whether caspase-1 in distinct cell types performs specific roles in inflammasome activity or if a single population is responsible for the release of multiple leaderless cytokines and alarmins.

To investigate caspase-1-mediated immunity against *T. gondii*, we used whole-body and cell-type-specific knockout mice. Using this approach, we report that caspase-1 is required for optimal parasite control, promoting CD4[+] T cell IFN-γ production and enhancing neutrophil and monocyte recruitment to the initial site of infection. In particular, we find that *Cx3cr1*-positive cells use caspase-1 to release IL-18, which reinforces IFN-γ production. On the other hand, monocyte and neutrophil responses were intact in these mice, which suggests that caspase-1 activity in additional cell types regulates these innate responses. Our studies demonstrate that in the context of *in vivo T. gondii* infection caspase-1 is utilized in distinct cell populations to carry out non-overlapping and specific downstream effects of caspase-1-mediated immunity to *T. gondii*.

## Results

### Caspase-1 mediates aspects of innate and adaptive immunity to T. gondii

The initial studies examining the role of inflammasomes in *T. gondii* infection were performed in double knockout mice, in which expression of both *Casp1* and *-11* were ablated and immunity against the parasite was impaired [19–21]. We hypothesized that the caspase-1 canonical inflammasome pathway would be sufficient to drive multiple aspects of protective immunity towards *T. gondii*. Therefore, we infected *Casp1* single knockout mice with 10 cysts of Me49 type II strain parasite by intraperitoneal (i.p) infection [31]. Then, we performed peritoneal lavage at eight days post-infection (8 dpi) and found that the parasite burden was 3.6-fold greater in *Casp1* deficient mice compared to WT controls (Fig 1A). Additionally, during chronic infection at six weeks post-infection (6 wpi), we found a 1.4-fold increase in parasite burden in the brains of *Casp1* KO mice relative to control mice (S1A Fig). This was accompanied with a 2-fold increase in tachyzoite-specific gene expression (S1B and S1C Fig). Surprisingly, *Casp1* and *Casp1/11* KO mice had similar long-term survival post-infection and *Casp1* KO mice were protected from infection induced weight loss compared to WT mice at 35 days post infection, but consistently had higher parasite burdens (S1D–S1E Fig). To test if the increased parasite burden at day 8 was due to an immunological defect, we performed ELISAs on the serum of WT and *Casp1* KO mice. We found that *Casp1* KO mice had a 25% reduction in IFN-γ levels compared to WT controls (Fig 1B) but had comparable IL-12 serum levels (Fig

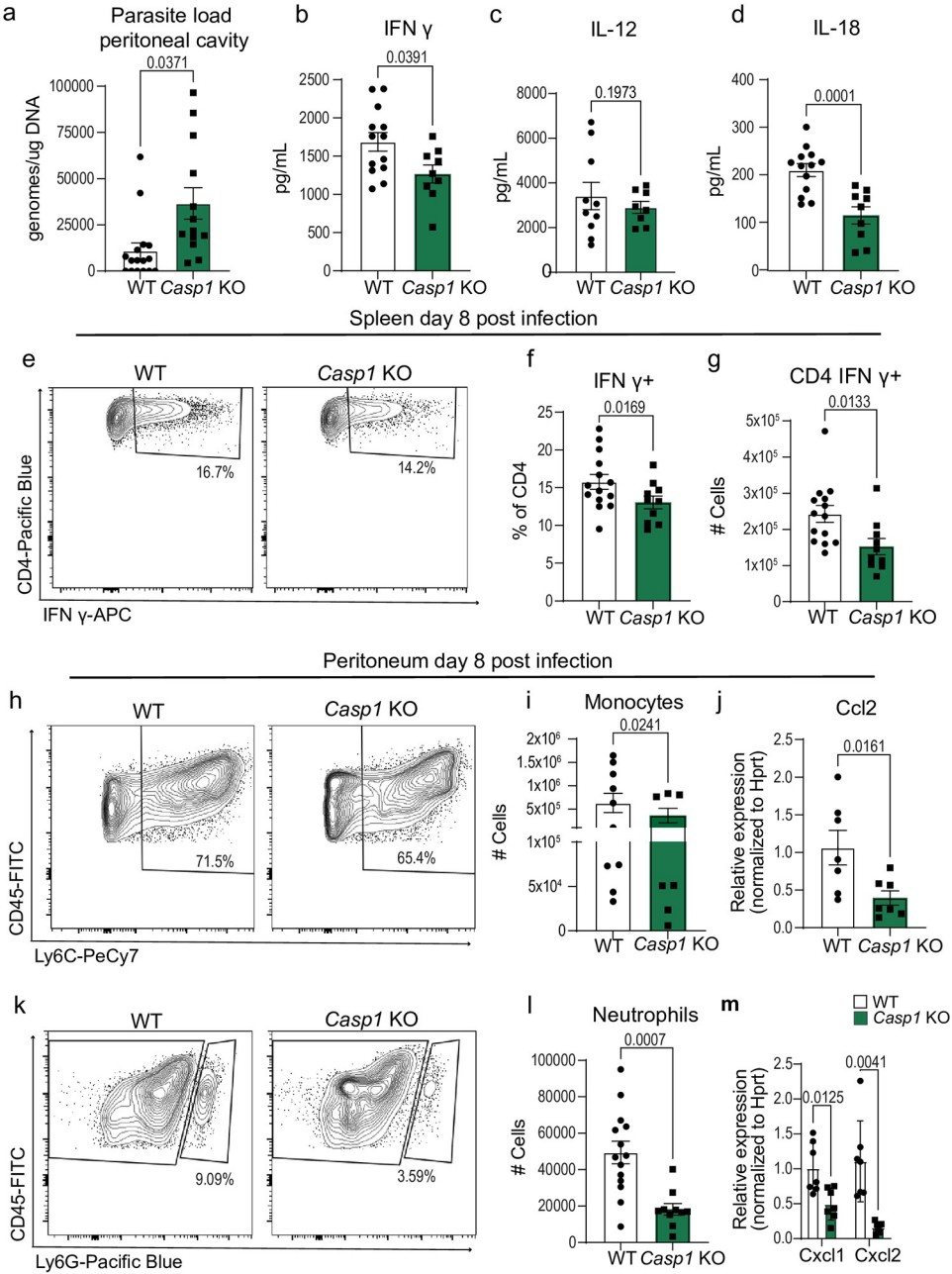

**Fig 1. Caspase-1 mediates innate and adaptive immune responses to T. gondii.** (a) qPCR analysis of *T. gondii* parasite load 8 days post-infection (8 dpi) in the peritoneum of wildtype WT C57BL/6 (n = 16) and *Casp1* Knockout (*Casp1* KO) (n = 13) mice, four experiments. (b-d) Serum cytokine levels 8 dpi, three experiments for (b) IFN-γ: WT (n = 13), *Casp1* KO (n = 9); (c) IL-12: WT (n = 10), *Casp1* KO (n = 8); and (d) IL-18: WT (n = 13), *Casp1* KO (n = 9). (e-g) Flow cytometry of CD3⁺ CD4⁺ IFN-γ⁺ T cells in spleen at 8 dpi, three experiments, WT (n = 14), *Casp1* KO (n = 10). (e) Representative flow cytometry plot. (f) Frequency of CD4⁺ T cells IFN-γ⁺. (g) Number of CD4⁺ T cells producing IFN-γ. (h) Representative flow cytometry plots of CD45⁺ CD11b⁺ Ly6C⁺ monocytes in the peritoneum 8dpi. (i) Number of CD45⁺ CD11b⁺ Ly6C⁺ monocytes in WT (n = 10), *Casp1* KO (n = 7), two experiments. (j) RT-qPCR analysis of peritoneal Ccl2 expression in WT (n = 7) and *Casp1* KO (n = 7), two experiments. (k) Representative flow cytometry plot of CD45⁺ CD11b⁺ Ly6G⁺ neutrophils in the peritoneum at 8 dpi. (l) Number of CD45⁺ CD11b⁺ Ly6G⁺ neutrophils in WT (n = 15), *Casp1* KO (n = 9), three experiments. (m) RT-qPCR analysis of peritoneal Cxcl1 and Cxcl2 expression in WT (n = 7) and *Casp1* KO (n = 7), two experiments. Data are mean ± s.e.m., p values by randomized-block ANOVA and post-hoc Tukey test (a-d, f-g, i-j, and l-m).

1C). On the other hand, we found that IL-18 serum levels in *Casp1* KO mice were nearly half of what was measured in WT controls (Fig 1D). We were unable to detect IL-1α or IL-1β in the serum at this timepoint, consistent with prior reports [20,21,32].

As IFN-γ is a major regulator of immunity to *T. gondii*, we then aimed to identify cell populations where IFN-γ production was impacted by *Casp1* deficiency. We harvested spleen and peritoneal exudate cells at 8 days post-infection and used spectral flow cytometry to identify IFN-γ-producing cell populations (S1F–S1Q Fig). We found fewer CD4$^+$ T cells that produced IFN-γ within the spleen in *Casp1* KO mice in comparison to WT controls, but the total number of activated T cells did not differ (Figs 1E–1G and S2A). Additionally, *Casp1* KO had no impact on splenic CD8$^+$ T cell's IFN-γ production but did result in more T-bet$^+$ CD8$^+$ T cells compared to WT controls (S2B and S2C Fig). CD4$^+$ T-bet expression was unchanged (S2C Fig). NK cells, and ILC1/NK T cell IFN-γ production was also unchanged (S2D and S2E Fig). In the peritoneum 8 dpi, in *Casp1* KO mice, no defect was seen in CD4$^+$ T cells or NK/ ILC1 and NK T cells (S2F–S2K Fig). Although, we did observe more activated CD8$^+$ T cells in the peritoneum of *Casp1* KO mice compared to WT controls (S2F Fig). Together these data highlight a defect in IFN-γ production specific to splenic CD4$^+$ T cells in mice lacking *Casp1*.

Beyond a defect in IFN-γ production, we observed a dramatic decrease in the number of neutrophils (CD11b$^+$Ly6G$^+$) and monocytes (CD11b$^+$ Ly6G$^-$ Ly6C$^+$) recruited to the peritoneum in *Casp1* KO mice compared to WT mice (Figs 1H–1M, S1R–S1U, and S2L and S2M). Corresponding to the decreased monocyte infiltration, in the peritoneum we found decreased levels of the mRNA for the monocyte chemoattractant, Ccl2, in the peritoneum of *Casp1* KO mice compared to WT (Fig 1J). In *Casp1* KO peritoneal lavage fluid mRNA levels of Cxcl1 and Cxcl2, major chemokines for neutrophil attraction, were also decreased in comparison to WT controls (Fig 1M). In line with peritoneal IFN-γ production being similar between WT and KO mice, we did not observe a difference in iNOS production from peritoneal monocytes in *Casp1* KO mice in comparison to WT controls (S2N and S2O Fig). To test whether the defects seen in *Casp1* KO mice occur in naïve mice, we harvested peritoneal myeloid cells and splenic T cells from naïve WT and *Casp1* KO mice. We did not observe any significant differences in cell populations or T cell cytokine production in naïve WT and *Casp1* KO mice (S2P–S2U Fig). Together these data demonstrate that *Casp1* deficiency leads to innate and adaptive immune defects that are associated with an inability to fully control parasite levels during acute *T. gondii* infection.

## IL-18 promotes CD4$^+$ T cell IFN-γ production but is dispensable for myeloid cell recruitment to the initial site of infection

A recent study by Clark et al. identified that IL-18 can promote CD4$^+$ T-cell IFN-γ production during *T. gondii* infection [30]. With IL-18 being a molecule released downstream of caspase-1, we were curious as to when during infection IL-18 is released. We sampled serum and peritoneal fluid (Per-Fluid) in naive mice (0 dpi) and at 3 and 6 days post infection (3 dpi, 6 dpi). We observed that both IL-18 and IFN-γ increased in levels from 3 dpi to 6 dpi (Fig 2A). Additionally, when we purified peritoneal exudate cells (PECs) at these time points and performed ex vivo cytokine release assays we observed that IL-18 was released at 6 dpi but not earlier (S3A Fig). Moreover, when we lysed purified PECs from these timepoints we did not detect significant differences in total IL-18 levels (S3B Fig). Next, we hypothesized that the decreased IL-18 serum levels in whole-body *Casp1* KO mice impaired CD4$^+$ T cell IFN-γ production. To test this hypothesis we infected *Il18* KO and WT control mice to gain insight on which immune responses downstream of caspase-1 activation were dependent on IL-18. Mice were infected with 10 cysts Me49 followed by analysis of peritoneal exudate cells and spleens at

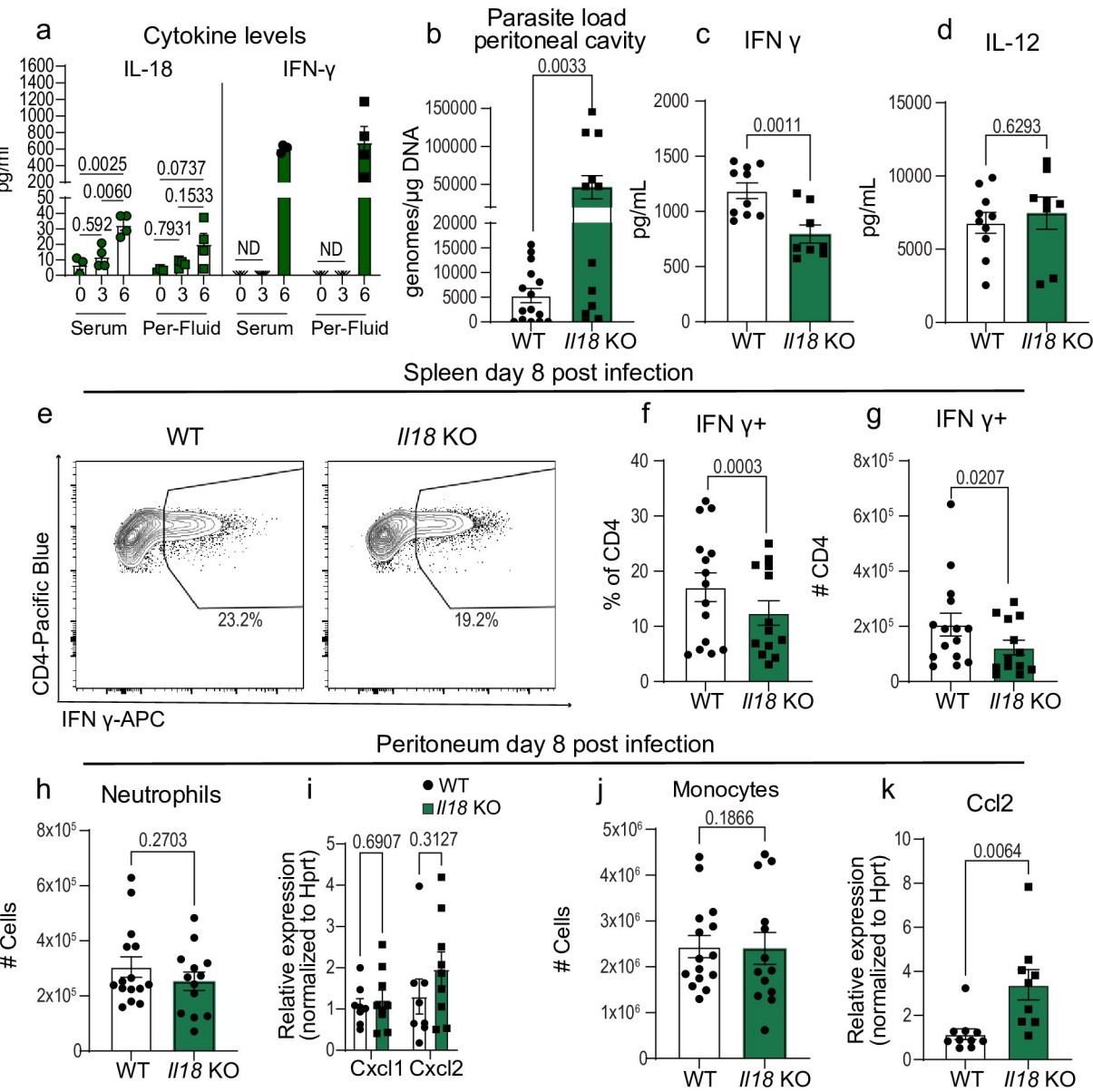

**Fig 2. IL-18 promotes CD4⁺ T cell IFN-γ production but is dispensable for myeloid cell recruitment to the initial site of infection.** (a) IL-18 and IFN-γ levels in serum and peritoneal fluid (Per-Fluid) at 0 dpi (n = 3), 3 dpi (n = 4), 6 dpi (n = 4). (b) qPCR analysis of *T. gondii* parasite load 8 days post-infection (8 dpi) in the peritoneum of wildtype (WT) C57BL/6 (n = 15) and *Il18* KO (n = 12) mice, three experiments. (c,d) Serum cytokine levels 8 dpi, two experiments for (c) IFN-γ: WT (n = 10), *Il18* KO (n = 8); (d) IL-12: WT (n = 10), *Il18* KO (n = 8). (e-g) Flow cytometry of CD3⁺ CD4⁺ IFN-γ⁺ T cells in spleen at 8 dpi, three experiments, WT (n = 15), *Il18* KO (n = 13). (e) Representative flow cytometry plot. (f) Frequency of CD4⁺ T cells IFN-γ⁺. (g) Number of CD4⁺ T cells producing IFN-γ. (h) Number of CD45⁺ CD11b⁺ Ly6G⁺ neutrophils in WT (n = 15), *Il18* KO (n = 13), three experiments. (i) RT-qPCR analysis of peritoneal Cxcl1 and Cxcl2 expression in WT (n = 7) and *Il-18* KO (n = 9), two experiments. (j) Number of CD45⁺ CD11b⁺ Ly6C⁺ monocytes in WT (n = 15), *Il18* KO (n = 13), three experiments. (k) RT-qPCR analysis of peritoneal Ccl2 expression in WT (n = 10) and *Il18* KO (n = 9), two experiments. Data are mean ± s.e.m., p values by one-way ANOVA with post-hoc Tukey test (a) or randomized-block ANOVA and post-hoc Tukey test (b-d and f-k).

eight days post-infection. Similar to *Casp1* deficient mice, *Il18* deficient mice displayed an increased parasite burden in the peritoneal cavity and decreased serum IFN-γ levels (Fig 2B and 2C). Likewise, *Il18* deficiency had no effect on serum IL-12 levels (Fig 2D). In the spleens of *Il18* deficient mice, IFN-γ production by CD4⁺ and CD8⁺ T cells was impaired (Figs 2E–2G

and S3C and S3D). Similarly, in the peritoneal cavity of *Il18* KO mice a defect in IFN-γ production by CD4$^+$ and CD8$^+$ T cells was observed (S3E–S3H Fig). These data suggest that IL-18 promotes CD4$^+$ T cell IFN-γ production during the acute phase of *T. gondii* infection and is consistent with results observed in *Casp1* deficient mice. On the other hand, when we assayed myeloid cell responses in the peritoneum, neutrophil and monocyte numbers were comparable between *Il18* deficient and WT controls (Figs 2H–2K and S3I–S3L). Together, these results suggest that IL-18 is essential for full parasite control and likely represents one arm of the caspase-1-dependent immune response during the acute stage of *T. gondii* infection.

Given the specificity of the IL-18 response, we next examined IL-18 receptor (IL-18R) expression to begin to understand why IFN-γ production is strongly impaired in splenic CD4$^+$ T cells in *Il18*-deficient mice at day 8 post-infection. We quantified IL-18R-positive CD4$^+$ and CD8$^+$ T cells in the spleen and peritoneal cavity at 8 days post-infection (S3M Fig). Approximately 30% of CD4$^+$ T cells in the spleen and nearly 80% of CD4$^+$ T cells in the peritoneum expressed IL-18R, while only a small fraction of CD8$^+$ T cells expressed the receptor (S3N and S3O Fig). To understand if IL-18R expression on T cells is infection-dependent, we compared IL-18R expression between naïve and infected mice. We found that few splenic T cells express IL-18R at baseline (S3P Fig). These data suggest that CD4$^+$ T cells are more responsive to IL-18, due to infection-induced upregulation of IL-18R.

## IL-1Rap mediates neutrophil recruitment but is dispensable for parasite control at the site of infection

IL-18 is one of many molecules that can be released downstream of caspase-1. To broadly test if the loss of other leaderless cytokine signaling could recapitulate whole-body *capase-1* deficiency during acute *T. gondii* infection, we utilized *IL-1 receptor accessory protein (Il1rap)* knockout mice. IL-1RAP is a necessary receptor subunit for IL-1, IL-33, and IL-36 receptor signaling. We infected *Il1rap* WT and KO mice with 10 cysts Me49 and performed peritoneal lavage at eight days post infection. We observed no difference in parasite burden between *Il1rap* KO and WT control mice at this site (Fig 3A). Furthermore, we observed no difference in IFN-γ serum levels or splenic CD4$^+$ T cell IFN-γ production (Fig 3B–3E). In the peritoneum, *Il1rap* KO mice recruited similar numbers of immune cells and had comparable monocyte responses to WT controls (Fig 3C and 3F–3H). However, there was a severe impairment in neutrophil recruitment to the site of infection in *Il1rap* KO mice (Fig 3I–3K). These results are in line with a previous report showing that IL-1R signaling is required for neutrophil recruitment during acute infection and is dispensable for IFN-γ responses [25]. Taken together, these data suggest that IL-1RAP signaling is critical for neutrophil responses but is dispensable for parasite control in the presence of IFN-γ and monocytes.

## Caspase-1 in Cx3cr1-expressing cells is necessary for parasite control and optimal production of IFN-γ by CD4$^+$ T cells

To begin to address how caspase-1 shapes immunity to *T. gondii*, we sought to identify which cells type(s) are involved. We aimed to identify the population responsible for caspase-1-mediated IFN-γ production and parasite control. To this end, we utilized a single-cell RNA sequencing dataset that examined gene expression in the spleen at day 14 post-infection with *T. gondii*. We focused on cell types that express IL-18, as the release of this cytokine is significantly decreased in *Casp1* deficient mice. At this timepoint, we confirmed that macrophages are a major source of *Il18* and *Casp1* expression (S4A and S4B Fig). We hypothesized that caspase-1 activity in macrophages would be necessary for parasite control, optimal cytokine production, and cellular recruitment to the peritoneal cavity. To test this hypothesis, we crossed

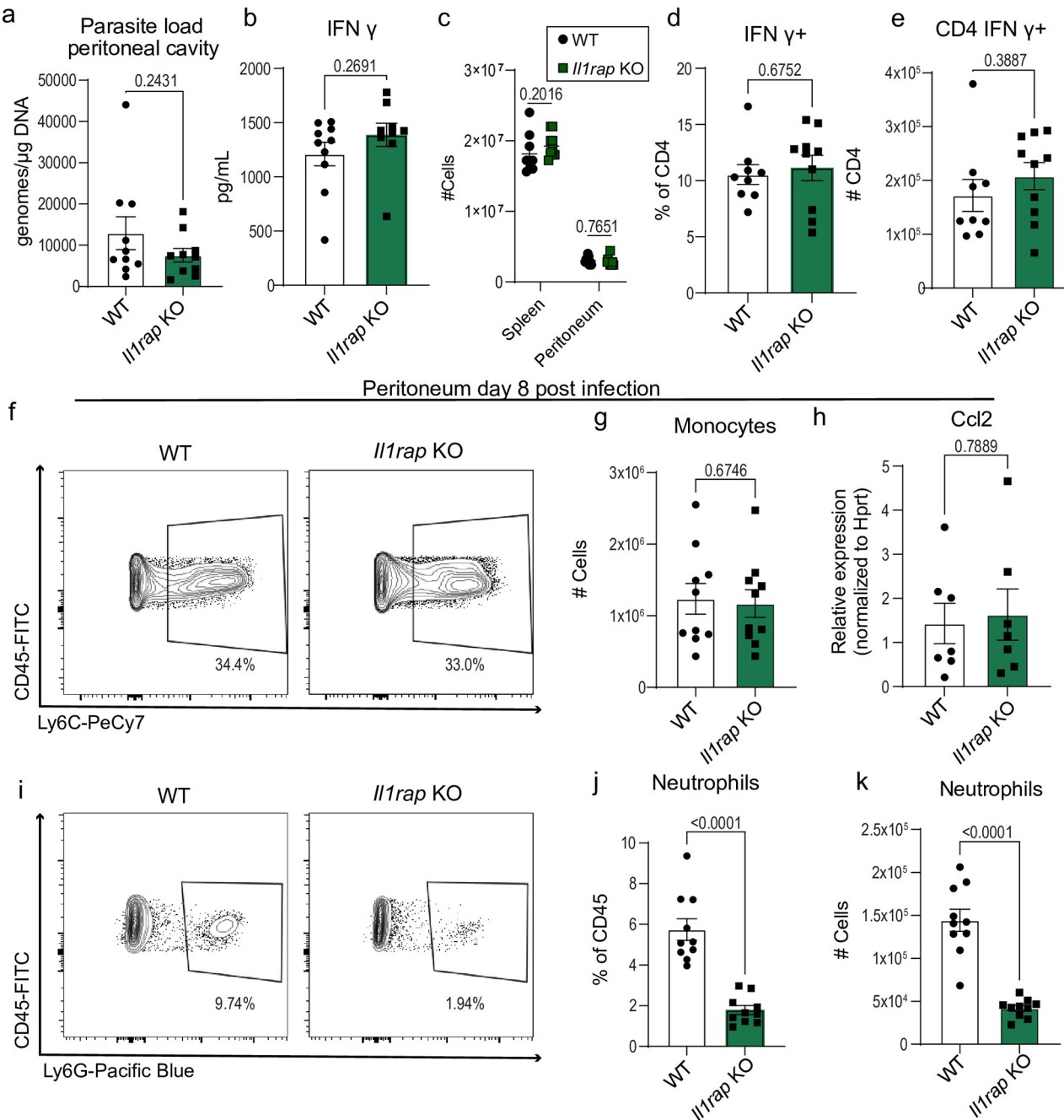

**Fig 3. IL-1RAP signaling mediates neutrophil recruitment to the sight of infection but is dispensable for parasite control during acute T. gondii infection.** (a) qPCR analysis of *T. gondii* parasite load 8 days post-infection (8 dpi) in the peritoneum of *Il1rap* KO and wildtype (WT) mice, C57BL/6 (*n* = 10) and *Il1rap* KO (*n* = 10) mice, two experiments. (b) Serum cytokine levels 8 dpi, two experiments for IFN-γ: WT (*n* = 10), *Il1rap* KO (*n* = 10). (c) Cell counts from spleen and peritoneum: WT (n = 9), *Il1rap* KO (n = 10). (d-e) Flow cytometry of CD3$^+$ CD4$^+$ IFN-γ$^+$ T cells in spleen at 8 dpi, two experiments, WT (*n* = 9), *Il1rap* KO (*n* = 10). (d) Frequency of CD4$^+$ T cells IFN-γ$^+$. (e) Number of CD4$^+$ T cells producing IFN-γ. (f) Representative flow cytometry plots of CD45$^+$ CD11b$^+$ Ly6C$^+$ monocytes in the peritoneum at 8dpi. (g) Number of CD45$^+$ CD11b$^+$ Ly6C$^+$ monocytes in WT (*n* = 10), *Il1rap* KO (*n* = 10), two experiments. (h) RT-qPCR analysis of peritoneal Ccl2 expression in WT (*n* = 7) and *Il1rap* KO (*n* = 7), two experiments. (i) Representative flow cytometry plot of CD45$^+$ CD11b$^+$ Ly6G$^+$ neutrophils in the peritoneum at 8 dpi. (j) Frequency and (k) Number of CD45$^+$ CD11b$^+$ Ly6G$^+$ neutrophils in WT (n = 10) and *Il1rap* KO (n = 10), two experiments. Data are mean ± s.e.m., p values by randomized-block ANOVA and post-hoc Tukey test (a-e, g-h, and j-k).

mice that constitutively express cre recombinase under the *Cx3cr1* promoter to a *Caspase-1*fl/fl background [33,34]. We infected *Cx3cr1*cre/+ x *Caspase* 1 fl/fl (*Cx3cr1*+ *Casp1* KO) and *Cx3cr1*+/+ x *Caspase* 1fl/fl (WT) littermate controls with 10 cysts Me49 and at eight days post infection we harvested tissues. In the peritoneum, we saw a ~3-fold increase in parasite burden in the *Cx3cr1*+ *Casp1* KO mice compared to control mice (Fig 4A), which was associated with a greater cyst burden in the brains of these mice at 6 weeks post-infection (6wpi) (S5A Fig). We did not observe a statistically significant difference in parasite burden in the spleen eight days post infection (S5B Fig). At day eight post-infection, we performed ELISAs on the serum and found that *Cx3cr1*+ *Casp1* KO mice had decreased IFN-γ levels, comparable IL-12 levels, and decreased IL-18 levels compared to WT mice (Fig 4B–4D). Additionally, we found comparable total levels of IL-18 in PECs but observed a significant defect in the percent of IL-18 released from PECs derived from *Cx3cr1*+ *Casp1* KO mice (S5C and S5D Fig). These data suggest that a *Cx3Cr1*+ cell population promotes IL-18 release and enhances IFN-γ production. Using spectral cytometry, we found that *Cx3cr1*+ *Casp1* KO mice had impaired production of IFN-γ specifically in splenic CD4+ T cells in response to *T. gondii* infection, but there were not differences found at baseline (Figs 4E–4G and S5E–S5H). In the peritoneal cavity, T cell responses and the recruitment of neutrophils and iNOS-producing monocytes were intact and comparable to WT levels (Figs 4H–4M and S5I–S5N). These data show that *Cx3cr1*+ *Casp1* KO mice exhibited a selective T cell defect and are nearly identical in phenotype to the *Il18* knockout mice.

## Cx3cr1-expressing cell populations in the peritoneal cavity undergo dynamic changes throughout the course of acute *T. gondii* infection

To track *Cx3cr1* expressing cells during acute *T. gondii* infection, we utilized *Cx3cr1* inducible reporter mice (Cx3cr1CreERT2/+Rosa26Ai6/Ai6). Mice were given tamoxifen chow for two weeks prior to infection then put on normal chow during infection. We observed that in naïve mice, 3dpi, and at 6dpi, the majority of *Cx3cr1* expressing cells are CD11b-positive, and only a small subset are CD11b-negative and express CD3 (S6A–S6G Fig). In naïve and 3dpi PECs the vast majority of *Cx3cr1*-expressing cells were macrophages (F480+ Ly6C-) (S6F Fig). However, by 6dpi there were significantly fewer *Cx3cr1*-expressing macrophages and the prominent *Cx3cr1*-expressing cell had characteristics of infiltrating monocytes (F480+ Ly6C+) (S6F Fig). We used fluorescence activated cell sorting (FACS) to isolate the *Cx3cr1*+ myeloid populations at 0 dpi, 3 dpi, and 6 dpi and performed RT-qPCR for Casp1, Il18, and Gbp2. We observed that at 0 dpi and 3 dpi these cells expressed low levels of Casp1 and do not seem equipped to undergo caspase-1-dependent IL-18 release. At 6 dpi all *Cx3cr1*+ myeloid populations expressed *Casp1* and *Il18* (S6H Fig). Seeing that Casp1 and Il18 expression increased at a similar time as the IFN-γ regulated gene Gbp2, we asked whether IFN-γ promotes caspase-1 dependent Il18 release in *Cx3cr1*+ cells. We utilized *Cx3cr1*CreERT2/WT *Stat1* fl/fl mice, which renders Cx3cr1 expressing cells incapable of responding to IFN-γ. *Stat1* deletion did not significantly impact *Cx3cr1*+ *Casp1*-peritoneal parasite burden, IL-18 cytokine levels, Casp1 and Il18 transcript expression, or percent and number of CD4+ and CD8+ T cells producing IFN-γ (S7A–S7G Fig). Stat1 deficiency did appear to affect Casp1 expression and is consistent with increased Casp1 expression at a timepoint when IFN-γ production and downstream IFN-γ regulated gene *Gbp2* is increased. Nevertheless, IFN-γ signaling on *Cx3cr1*+ cells was not required to promote IL-18 release during infection. Together these data suggests that during acute *T. gondii* infection *Cx3cr1*+ cells are initially primarily myeloid and dynamically change from 3dpi to 6dpi.

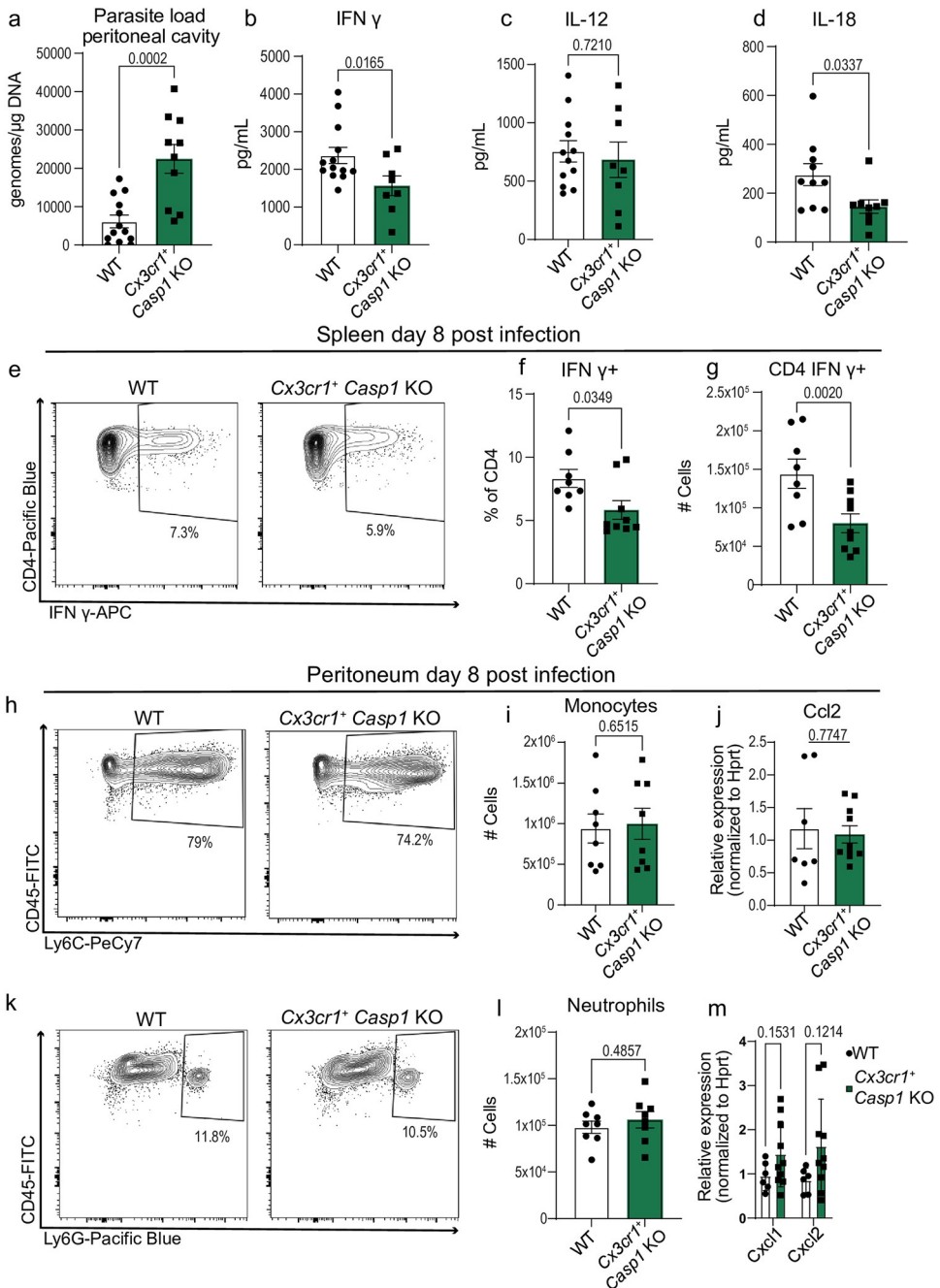

**Fig 4. Caspase-1 in Cx3cr1-expressing cells is necessary for parasite control and optimal CD4$^+$T cell production of IFN-γ.** (a) qPCR analysis of *T. gondii* parasite load 8 days post-infection (8 dpi) in the peritoneum of *Cx3cr1$^{+/+}$ Caspase 1$^{fl/fl}$* (WT) (n = 13) and *Cx3cr1$^{Cre/+}$ Caspase 1$^{fl/fl}$* (*Cx3cr1$^+$ Casp1* KO) (n = 10) mice, three experiments. (b-d) Serum cytokine levels 8 dpi, three experiments for (b) IFN-γ: WT (n = 13), *Cx3cr1$^+$ Casp1* KO (n = 8); (c) IL-12: WT (n = 12), *Cx3cr1$^+$ Casp1* KO (n = 8); and (d) IL-18: WT (n = 10), *Cx3cr1$^+$ Casp1* KO (n = 9). (e-g) Flow cytometry of CD3$^+$ CD4$^+$ IFN-γ$^+$ T cells in spleen at 8 dpi, two experiments, WT (n = 8), *Cx3cr1$^+$ Casp1* KO (n = 9). (e) Representative flow cytometry plot. (f) Frequency of IFN-γ-producing CD4$^+$ T cells (g) Number of CD4$^+$ T cells producing IFN-γ. (h) Representative flow cytometry plots of CD45$^+$ CD11b$^+$ Ly6C$^+$ monocytes in the peritoneum 8 dpi. (i) Number of CD45$^+$ CD11b$^+$ Ly6C$^+$ monocytes in WT (n = 8) and *Cx3cr1$^+$ Casp1* KO (n = 8), two experiments. (j) RT-qPCR analysis of peritoneal Ccl2 expression in WT (n = 7) and *Cx3cr1$^+$ Casp1* KO (n = 10), two experiments. (k) Representative flow cytometry plot of CD45$^+$ CD11b$^+$ Ly6G$^+$ neutrophils in the peritoneum at 8 dpi. (l) Number of CD45$^+$ CD11b$^+$ Ly6G$^+$ neutrophils in WT (n = 8), *Cx3cr1$^+$ Casp1* KO (n = 9), two experiments. (m) RT-qPCR analysis of peritoneal Cxcl1 and Cxcl2 expression in WT (n = 6) and *Cx3cr1$^+$ Casp1* KO (n = 10), two experiments. Data are mean ± s.e.m. p values by randomized-block ANOVA and post-hoc Tukey test (a-d, f-g, i-j, and l-m).

### Recombinant IL-18 administration rescues CD4[+] T cell IFN-γ production and parasite control in Cx3cr1[+] caspase-1 deficient mice

*Cx3cr1[+] Casp1* KO mice had decreased serum IL-18 levels and phenocopied *IL-18* KO mice with both having selective impairment in CD4[+] T cell IFN-γ production. Thus, we hypothesized that the decreased IL-18 levels in *Cx3cr1[+] Casp1* KO mice leads to decreased CD4[+] T cell IFN-γ production and parasite control. To test this hypothesis, we performed a rescue experiment, where we administered recombinant IL-18 (rIL-18) at 10µg/kg or PBS to *Cx3cr1[+] Casp1* deficient mice at 1 hour post-infection and on days 1, 3, 5, and 7 post-infection (Fig 5A). We confirmed that the rIL-18 treatment restored IL-18 serum levels (Fig 5B). As expected, parasite burden was greater in *Cx3cr1[+] Casp1* KO mice that received PBS treatment compared to WT controls. However, in *Cx3cr1[+] Casp1* KO mice that received rIL-18, parasite burden was comparable to levels seen in WT mice (Fig 5C). Furthermore, we found that rIL-18 treatment was able to partially rescue IFN-γ serum levels while leaving IL-12 levels unchanged (Fig 5D and 5E). Together, these data demonstrate that administering rIL-18 in *Cx3cr1[+] Casp1* KO mice was sufficient to restore IFN-γ production and ultimately recuperate parasitic control. We hypothesized that the recovery of IFN-γ serum levels in mice that received rIL-18 administration would coincide with a recovery in frequency and number of splenic CD4[+] T cells making IFN-γ. Indeed, when flow cytometry was performed on splenic cells at 8 dpi, we found that rIL-18 was able to restore IFN-γ production to splenic CD4[+] T cells in *Cx3cr1[+] Casp1* KO mice (Fig 5F–5H). Thus, administration of rIL-18 was sufficient to rescue CD4[+] T cell IFN-γ production and parasite control in mice lacking caspase-1 production in *Cx3cr1[+]* expressing cells. Taken together, these results highlight the importance of caspase-1 in *Cx3cr1*-positive populations to promote IL-18 release. These studies demonstrate the varied contributions of caspase-1-mediated pathways during an infection. One arm of the caspase-1 response mediates IL-18 release to impact T cell cytokine production and pathogen control, while other arms stimulate signaling (including through IL-1RAP), to mediate cellular recruitment to the site of infection.

## Discussion

Prior studies on caspase-1/11 during *T. gondii* infection have revealed the capacity of inflammasomes to release various molecules *in vivo* and *in vitro* [19–21,24,35]. However, prior to the current study, the necessity of caspase-1 rather than caspase1/11 and the specific cell types utilizing caspase-1 and the immunological responses triggered by caspase-1 activity *in vivo* during *T. gondii* infection were unknown. A recent study by Sateriale et. al, 2021 demonstrated that *Casp1* alone is sufficient for controlling cryptosporidiosis raising the possibility that *Casp1* could be sufficient for the control of other parasitic infections [36]. Additionally, early studies on caspase-1 have employed *in vitro* cultures utilizing a single cell type. The development of caspase-1[fl/fl] mice has allowed for a more detailed exploration of caspase-1 activity *in vivo* [34]. This study presents evidence indicating that distinct cell populations use caspase-1 to support different facets of immunity to *T. gondii*. Specifically, *Cx3cr1*-positive populations utilize caspase-1 to promote IL-18 release, which facilitates CD4[+] T cell IFN-γ production [30]. Although, loss of *Casp1* from *Cx3cr1*-positive populations did not lead to a complete loss of IL-18, suggesting other cell populations can release this molecule [20,21]. In contrast, another population(s) leverages caspase-1 for neutrophil recruitment through IL-1RAP signaling and monocyte recruitment via induction of the chemokine CCL2.

We identified a caspase-1/IL-1RAP signaling axis as a necessary component of neutrophil recruitment to the site of infection. The role of neutrophils during acute *T. gondii* has been an active area of investigation. A previous report using an anti-Gr-1 antibody to deplete

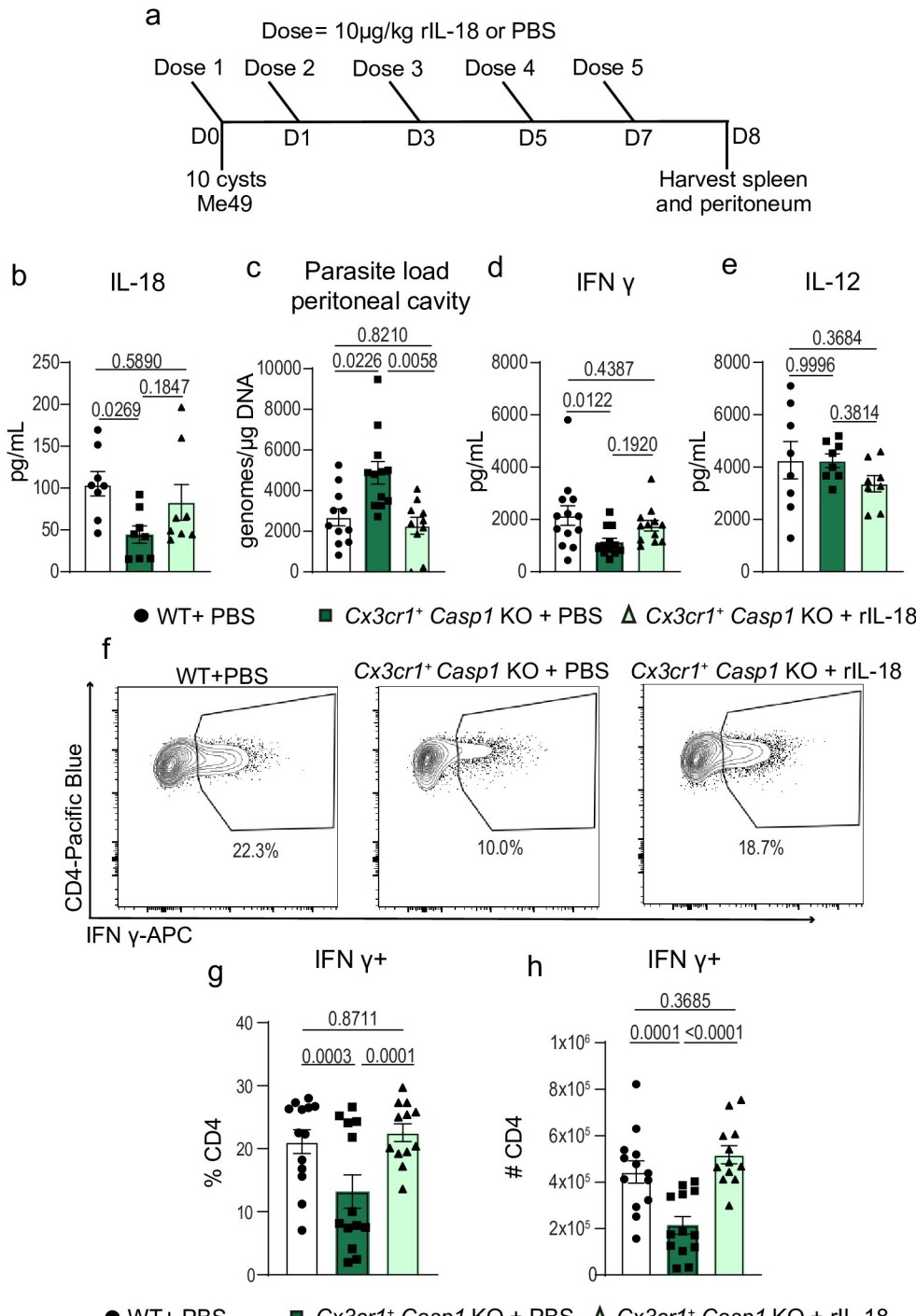

**Fig 5. Recombinant IL-18 administration rescues CD4⁺ T cell IFN-γ production and parasite control in Cx3cr1⁺ caspase-1 deficient mice.** (a) Schematic of experimental design. (b) Serum cytokine levels of IL-18, two experiments, WT + PBS (*n* = 8), *Cx3cr1⁺ Casp1* KO + PBS (*n* = 8), and *Cx3cr1⁺Casp1* KO + rIL-18 (*n* = 8). (c) qPCR analysis of parasite load in peritoneal cavity, three experiments, WT + PBS (*n* = 11), *Cx3cr1⁺ Casp1* KO + PBS (*n* = 12), and *Cx3cr1⁺ Casp1* KO + rIL-18 (*n* = 10). (d) Serum cytokine levels of IFN-γ, three experiments, WT + PBS (*n* = 13), *Cx3cr1⁺ Casp1* KO + PBS (*n* = 12), and *Cx3cr1⁺ Casp1* KO + rIL-18 (*n* = 12). (e) Serum cytokine levels of IL-12, two experiments, WT + PBS (*n* = 8), *Cx3cr1⁺Casp1* KO + PBS (*n* = 8), and *Cx3cr1⁺ Casp1* KO + rIL-18 (*n* = 8). (f-h) Flow cytometry of CD3⁺ CD4⁺ IFN-γ⁺ T cells in spleen at 8 dpi, three experiments, WT + PBS (n = 13), *Cx3cr1⁺ Casp1* KO + PBS (n = 12), and *Cx3cr1⁺ Casp1* KO + rIL-18 (n = 12). (f) Representative flow cytometry plot. (g) Frequency of CD4⁺ T cells IFN-γ⁺. (h) Number of CD4⁺ T cells producing IFN-γ. Data are presented as mean ± s.e.m., p values by randomized-block ANOVA and post-hoc Tukey test (b-e, g-h).

neutrophils suggests that neutrophils are critical for parasite control [37]. Anti-Gr-1 is not specific to neutrophils and results in the depletion of inflammatory monocytes, making it difficult to interpret the importance of neutrophils during acute *T. gondii* infection. Interestingly, a recent *in vitro* study showed that *T. gondii*-induced neutrophil extracellular trap (NET) formation promoted the migration and cytokine response of T cells, suggesting that neutrophils can promote a more robust adaptive immune response [38]. A third study using an antibody targeting Ly6G to obtain a more selective depletion of neutrophils found no role for neutrophils in parasite control or IFN-γ levels in orally infected mice [39]. Similarly, our study using *Il1rap* KO mice which only had a defect in neutrophil recruitment to the site of infection suggests that neutrophils are not necessary for *T. gondii* control or IFN-γ production. Interestingly, both IL-33 signaling (ST2) and IL-1 signaling (IL-1R1) utilize IL-1RAP and have been associated with neutrophil recruitment during acute *T. gondii* infection [25,40]. In both of these studies the knockout of each respective receptor brought neutrophil levels drastically down. Thus, although neutrophil recruitment is regulated through caspase-1 and IL-1RAP signaling, this specific immunological response to *T. gondii* appears to be dispensable for parasite control and is in agreement with studies that used anti-Ly6G antibodies to deplete neutrophils. Whether caspase-1 mediates the release of IL-33, IL-1β, and/or IL-1α to promote neutrophil recruitment remains unresolved. Caspase-1 cleavage of IL-33 is suggested to inactive the cytokine [41] but the mechanism of IL-33 release is still under investigation [42]. Recent studies have identified that IL-33 is released from cells via gasdermin D pore formation, but the protease activating gasdermin D varied with the stimulus used to induce IL-33 release [43,44]. Thus, whether caspase-1 liberates IL-33 or inactivates IL-33 during *T. gondii* infection is of interest.

Our study on *Il1rap* KO mice not only provided insight the role of neutrophils during *T. gondii* infection but also on monocyte recruitment. As both *Il18* KO and *Il1rap* KO mice had no defect in monocyte recruitment, our data suggests an additional signal is promoting monocyte recruitment to the site of *T. gondii* infection. Our study aligns with a recent report describing how monocytes are recruited to the site of infection. Specifically, Safronova, et al. recently described that the alarmin S100a11 mediates *Ccl2* expression and monocyte recruitment to the peritoneal cavity following infection. In accordance, they found that S100a11 release was caspase-1/11-dependent, both *in vitro* and *in vivo* [24]. Together with our study, this suggests that a population(s) other than *Cx3cr1*[+] cells is driving caspase-1/S100a11-/CCL2-dependent monocyte recruitment.

Our study is also in accordance with a recent study from Clark et. al., which describes that IL-18 and its regulation impacts the ability of CD4[+] T cells to make IFN-γ [30]. Although, initially we were surprised to see that the NK cell IFN-γ response is intact in *Casp1* and *Il18* KO mice, López-Yglesias, et. al. also showed that *Casp1/11* deficiency had no impact on NK cell IFN-γ responses. One likely explanation is that the timing of IL-18 interaction with NK cells is critical. It has been observed that if an NK cell interacts with IL-12 before interacting with IL-18, IL-18 will not impact NK cell IFN-γ production [45]. During *T. gondii* infection IL-12 levels are detected and increase before IL-18 levels [46–48], thus NK cells may encounter IL-12 well before IL-18, explaining why IL-18 has minimal impact on NK cell activation. We were also surprised by the specificity of the IFN-γ defects in splenic CD4[+] T cells and not CD8[+] T cells in *Casp1* and *Cx3cr1*[+] *Casp1* knockouts. Additionally, these results can be explained by the preferential upregulation of IL-18R on CD4[+] T cells by day 8 post-infection. Interestingly, the report by Clark et. al. demonstrates that CD8[+] T cells upregulate IL-18R expression at day 10 post-infection [30]. Thus, the expression of IL-18R on CD8[+] T cells is tightly regulated by an unknown cue. Similarly, we found that NK cell/ILC1 cell population expression of IL-18R is absent at baseline and present at day eight post-infection, whereas, in accordance with Clark et. al., the NK T cell population expressed IL-18R at baseline and at day eight [30].

Interestingly, our study and the study from Clark et. al. showed that exogenous addition of IL-18 did not impact NK cell IFN-γ production, but Clark et. al. showed that when a recombinant IL-18 resistant to inhibition by IL-18 binding protein was exogenously added, NK cell responses and IFN-γ production were enhanced. These data suggest that the response of NK cells to IL-18 is highly regulated. We observed that the peritoneum did not phenocopy IFN-γ defects seen in the spleen. This difference in response may be due to the dynamic and evolving nature of IFN-γ production at this site of infection, suggesting 8 dpi may not be the optimal time where caspase-1 impacts peritoneal IFN-γ production or alternatively caspase-1 is not necessary for peritoneal IFN-γ production. Additionally, *Casp1* KO mice and *Cx3cr1*+ *Casp1* KO had detectable amounts of IL-18 in their serum but still had a similar reduction in IFN-γ serum levels as *Il18* KO mice [20,21]. This observation suggests a partial reduction in IL-18 could account for its maximal effect on splenic CD4+ T cell but not peritoneal CD4+ T cell IFN-γ production. Overall, these data suggest that IL-18 release and IL-18R expression are exquisitely controlled and ultimately influence the quality of CD4+ T cell production of IFN-γ.

While the current study has identified the importance of caspase-1 activity in specific cell populations, the signal(s) that leads to inflammasome activation remain unclear. Using Cx3cr1$^{CreERT2/WT}$ Stat1$^{fl/fl}$ mice, we did not observe a dependence on IFN-γ signaling for IL-18 expression or release. This result suggests that the cell type that releases IL-18 during infection does not need to respond to IFN-γ to express or release IL-18. Although, as multiple inflammasome sensors have shown to be triggered by *T. gondii*, it is possible that in the *Cx3cr1*-positive and other populations, caspase-1 is activated by distinct mechanisms and sensors. *In vitro*, both direct *T. gondii* infection and extracellular signaling cascades have been proposed to initiate the inflammasome [19,28,49,50]. Cell-intrinsic signals and cellular stress are also possible modes of inflammasome activation and differences in microbiota may play a role in inflammasome activation and its downstream impacts [51,52]. Although, during *in vivo T. gondii* infection, it is unclear whether direct invasion of the parasite and/or an infection-associated trigger is necessary for caspase-1 activation. A recent study by Wang, et. al., identified that in bone marrow derived macrophages (BMDMs) from the Lewis rat, NLRP1-dependent pyroptosis is mediated by direct infection and three parasite dense granule proteins GRA35, 42, and 43 [28]. In the human monocyte cell line (THP-1) cell cultures infected with *T. gondii*, the AIM2 inflammasome sensor has been shown to be activated by *T. gondii* DNA that has been liberated by guanylate binding protein (GBP) activity [49,53]. A third study using primary mouse peritoneal macrophages and BMDMs identified that NLRP3 was activated by extracellular ATP signaling released from parasite infected cells [50]. Our data along with these *in vitro* experiments raise the question of whether certain cell types can activate multiple inflammasome sensors and release multiple leaderless cytokines or DAMPs. In the single cell data set we analyzed, the splenic macrophage population containing IL-18 expressed AIM2 more than other inflammasome sensors [30]. We did not observe prominent expression of NLRP3 within the splenic macrophage population which is consistent with work from López-Yglesias et. al., which showed that NLRP3 does not impact CD4+ T cell IFN-γ production during acute *T. gondii* infection [21]. NLRP1 which has been shown to confer resistance to *T. gondii* infection in humans and rats [19,22,35] and has been reported to modulate IL-18 release in mice [20], was expressed in only a small percentage of splenic IL-18-expressing cells. Ultimately, whether one sensor or multiple sensors contribute to the distinct caspase-1 activities remains an open question.

In addition, the responses downstream of inflammasome activity are influenced by cellular expression of leaderless cytokines, leaderless cytokine receptors, and the chemokines a receptive cell can produce. Furthermore, the spatial location of these interactions likely influences the effects of inflammasome activity. For instance, IL-1α and IL-1β are regulated by a soluble

receptor antagonist (IL-1R antagonist) which competes with surface receptors. Thus, efficient IL-1 signaling may require local proximity of a IL-1R1-expressing cell [54]. Similarly, IL-18 is regulated by IL-18BP, making the proximity of IL-18 release near responding CD4$^+$ T cells critical for optimal IFN-γ production. To parse out this complex and specific interplay, high-dimensional spatially-resolved data sets will be of value. In this study, we were able to use a previously published single-cell RNA sequencing dataset to aid in the identification of a putative cell population capable of activating caspase-1 and releasing IL-18. The use of spatial transcriptomics and its ability to identify cells harboring leaderless cytokines and identify proximal cells able to respond to the cytokine will be advantageous for future studies on the orchestration of local inflammasome responses. The current study also demonstrates that distinct cell types use caspase-1 to impact innate and adaptive immune responses, which may provide an advantage to the host. The separation of the potent effects of caspase-1 activity across multiple cell types with differential inflammasome activation capacity likely acts as a checkpoint to prevent excess inflammation. Having a layered regulation of inflammasome activation *in vivo* allows for a broad range of activators to elicit specific and non-redundant responses.

As our study focused on the role of caspase-1 during acute *T. gondii* infection, it is unclear if a similar paradigm of cell-type specificity occurs in the brain during chronic *T. gondii* infection. As the brain contains long-lived cells which do not regenerate, the control of inflammation in this tissue is critical. A layered regulation of caspase-1 activity in the CNS would be advantageous for controlling *T. gondii* without promoting excess deleterious inflammation. A recent study by our group [25] identified that microglia, a *Cx3cr1*-positive cell population, releases IL-1α to promote myeloid cell recruitment into the infected brain and control parasite. This contrasts with acute infection where a population(s) other than *Cx3cr1*-positive cells contributes to myeloid recruitment. This highlights a significant difference in caspase-1 biology in the infected CNS and raises the important question of whether the CNS contains multiple cell types capable of activating caspase-1 to carry out unique aspects of immunity. Given that *Casp1* and *Casp1/11* deficient mice have higher parasite burdens in the CNS, it suggests that inflammasome activity is necessary in the brain and likely regulates local immune responses.

## Methods

### Ethics statement

All procedures involving animal care and use were approved by and conducted in accordance with the University of Virginia's Institutional Animal Care and Use Committee (IACUC) under protocol number 3968.

### Animals and treatments

WT (C57BL/6J), *Caspase-1*$^{-/-}$ (#32662) [31], *Caspase1/11*$^{-/-}$ (#016621) [55], *Cx3Cr1*$^{cre/cre}$ (#025524) [33], *Il18*$^{-/-}$ (#004130) [56] *Il1rap*$^{-/-}$ (#003284) [57] Cx3cr1$^{CreERT2/CreERT2}$ (#020940) [33] and ROSA26$^{Ai6/Ai6}$ (007906) strains were obtained from The Jackson Laboratory and maintained within our animal facility. Caspase-1$^{fl/fl}$ mice were provided by R. Flavell [34]. *Cx3Cr1*$^{cre/cre}$ mice were cross-bred with Caspase-1$^{fl/fl}$ mice to generate *Cx3Cr1*$^{cre/+}$ x Caspase-1$^{fl/fl}$. *Stat1*$^{fl/fl}$ mice (provided by Lothar Hennighausen, NIH) crossed to *Cx3Cr1*$^{cre/cre}$ and ROSA26$^{Ai6/Ai6}$ to generate *Stat1*$^{fl/fl}$ x Cx3cr1$^{CreERT2/WT}$ x ROSA26$^{Ai6/Ai6}$ (*Cx3cr1*$^+$ *Stat1* KO) mice [58]. The type II *T. gondii* strain Me49 was maintained in Swiss Webster mice (#024, Charles River Laboratories) and passaged through CBA/J mice (Jackson Laboratory). For experiments, tissue cysts were collected from chronically infected (>4 weeks) CBA/J mice. 7–10-week-old age and sex-matched mice were injected intraperitoneally with 10 cysts of Me49

in 200 μL of 1X PBS. Sham-infected mice were injected with an equal volume of 1X PBS. For rIL-18 experiments, 10 μg/kg or PBS of equivalent volume was given to mice on day 0, 1, 3, 5, and 7 post-infection with *T. gondii*. Mice used for endpoint studies were euthanized if they showed weight loss greater than 20% of their pre-infection bodyweight. For *Cx3cr1*[+] tracking experiments, mice were given tamoxifen chow (Envigo Teklad #TD.130858) for two weeks prior to analysis.

### *T. gondii* qPCR

Parasite genomic DNA was isolated from mouse peritoneal exudate cells and whole brain using the Isolate II Genomic DNA Kit (Bioline, BIO-52067). Prior to isolation, brain was first homogenized in 1X PBS using the Omni TH tissue homogenizer (Omni International). Amplification of *T. gondii* 529 bp repeat region using the SensiFAST Probe No-ROX Kit (Bioline, BIO-86005) and CFX384 Real-Time System (Bio-Rad) was performed as previously described [59] Tissue DNA (500 ng or 1μg) was loaded into each reaction. *T. gondii* isolated from human foreskin fibroblasts was used to make a serial standard curve from 3 to 300,000 genome copies and determine the number of *T. gondii* genomes per μg of tissue DNA.

### Tissue processing

Immediately after sacrifice a peritoneal lavage was performed with 3 mL of 1X PBS. Peritoneal material was pelleted, resuspended in 700 μL complete RPMI media (cRPMI; 10% FBS [Gibco], 1% penicillin/streptomycin [Gibco], 1% sodium pyruvate [Gibco], 1% non-essential amino acids [Gibco], and 0.1% 2-Mercaptoethanol [Life Technologies]). Subsequently, blood was collected from the superior vena cava, allowed to clot, and then spun down at 15 x *g* for 25 minutes. Serum was collected and stored at -70˚ C until further use. Then, transcardiac perfusion with 20 mL cold 1X PBS was performed. Spleens were taken and placed in 2 mL complete RPMI media. Spleens were then mechanically passed through a 40 μm filter (Corning, Ref 352340), spun down at 6,000 rpm for 3 min, and resuspended in red blood cell lysis buffer (0.16 M $NH_4Cl$) for 2 minutes. The cells were then washed with cRPMI, pelleted, and resuspended in 5 mL cRPMI. For intracellular cytokine staining, cells were resuspended in 50 μL cRPMI with Brefeldin A (Selleckchem Cat#S7046) for 5 hours at 37˚ C before staining.

### Flow cytometry and cell sorting

For myeloid cell panels, single cell suspensions were resuspended in Fc Block, made in FACS buffer (1X PBS, 0.2% BSA, and 2 mM EDTA) with 0.1 μg/ml 2.4G2 Ab (BioXCell, Cat#-CUS-HB-197) and 0.1% rat gamma globulin (Jackson Immunoresearch, Cat#012-000-002) for 10 minutes. Cells were stained for surface markers and eBioscience fixable live/dead viability dye efluor506 (1:800 Cat#65-0866-14) for 30 min at 4˚ C. Cells were then washed twice with 100 μL FACS buffer. For intracellular staining, cells were fixed with fixation/permeabilization solution (eBioscience, 00-5123-43 and 00-5223-56) for 15 min at room temperature. Cells were then washed twice with 100 μL permeabilization buffer (eBioscience, 00-8333-56) and stained for intracellular markers in 1X perm buffer for 20 min at 4˚ C and subsequently washed with 1X perm buffer. Finally, cells were resuspended in 200 μL FACS buffer and acquired using a 3 laser Cytek Aurora Flow Cytometry System. Data was analyzed using FlowJo software v10.9.0. For myeloid panels the following antibodies at 1:200 were used: CD45-FITC (eBioscience, Cat#11-0451-82), CD11b-APC-efluor780 (eBioscience, Cat#47-0112-82), CD11c-PerCP-Cyanine5.5 (eBioscience, Cat#45-0114-82), iNOS-APC (eBioscience, Cat#17-5920-82), Ly6G-Pacific Blue (Stemcell Technologies, Cat#60031PB), Ly6C-Pe-Cy7 (Biolegend, Cat#128018), F480-PerCP-Cyanine7 (Biolegend, Cat# 123113). For T cell and

cytokine panels the following antibodies at 1:200 were used: CD4-efluor450 (eBioscience, Cat#2114200), CD3-FITC (eBioscience, Cat#2159105), CD8α-PerCP-Cyanine5.5 (eBioscience, Cat#2410098), T-bet-PE-Cyanine7 (eBioscience, Cat#2410093), CD44-APC-efluor780 (eBioscience, Cat#2373711), IFN gamma-APC (eBioscience, Cat#2175632), CD218α (IL-18r)-PE (BioLegend, Cat#157903), CD62L-BV650 (BioLegend, Cat#104453), NK1.1-Super Bright 780 (eBioscience, Cat#78-5941-82).

For cell sorting Cx3cr1$^{\text{CreERT2/+}}$Rosa26$^{\text{Ai6/Ai6}}$ mice were used. After surface staining with F480 and Ly6C, cells were pre-gated on live, and Zsgreen$^{+}$ then sorted 4 ways based on F480 and Ly6C expression using a Sony MA900 and a 100 μM chip at the UVA flow cytometry core facility. Cells were collected in Trizol.

## ELISA and *ex vivo* cytokine release assay

Serum, peritoneal fluid, and media cytokine levels were detected according to R&D systems instructions using either DuoSet ELISA kits (IFN-γ, Cat# DY485-05) (IL-18, Cat# DY7625-05) with the DuoSet Ancillary Reagent Kit 2 (Cat# DY008B) or Quantikine ELISA (IL-12, Cat#M1240).

For ex vivo release assay, PECs were isolated and counted. 150,000 cells were plated into individual wells of a 96-well plate (Falcon, Cat#353077) and incubated in cRPMI for 16 hours. Supernatant was collected from wells untreated (released cytokine) or treated with 0.1% triton x100 (Sigma-Aldrich Cat#028K0011)(total cytokine). 100 μL of collected supernatant was used for ELISA.

## RT-qPCR

Peritoneal exudate cells were homogenized in Trizol. RNA was extracted according to the manufacturer's (Invitrogen) protocol. cDNA was then generated using a High-Capacity Reverse Transcription Kit (Applied Biosystems). Quantitative PCR was performed using 2X Taq based Master Mix (Bioline) and Taq Man gene expression assays (Applied Biosystems). Samples were run on a CFX384 Real-Time System thermocycler (Bio-Rad Laboratories). Genes were normalized to murine *Hprt*. The $2^{(-\Delta\Delta CT)}$ method was used to report relative expression [60]. The following ThermoFisher mouse gene probes were used: Hprt (Mm00446968_m1), *Ccl2* (Mm00441242_m1), *Cxcl1* (Mm04207460_m1), *Cxcl2* (Mm00436450_m1), Gbp2 (Mm 00494576_g1), Il18 (Mm00434226_m1). IDT gene probes were used: Ifngr1 (Hs.PT.58.20918191) and Casp1 (Mm.PT.58.8975671). Custom *T. gondii* RNA probes were used as previously reported [58].

## Single cell sequencing analysis

The publicly available count matrix from GSE207173 (Clark et al., 2023) was imported into a Python environment and processed using Scanpy [30]. For quality thresholds, cells with less than 1k counts, cells greater than 25k counts, or cells containing greater than 20% mitochondrial RNA content were filtered and excluded from the analysis. The data matrix was normalized and logarithmized, and PCA was used for dimensionality reduction using the ARPACK wrapper. Unsupervised clustering was performed using the leiden algorithm, with a resolution set to 0.6.

## Statistical analysis

All data was graphed in GraphPad Prism 9. To account for the biological variability between infections, data from experiment replicates was analyzed in R using a randomized block

ANOVA. This analysis models experimental groups as a fixed effect and experimental day as a random effect [61]. Outliers were identified and removed using ROUTs method with a Q value of 1 [62]. *p* values are indicated, with ns = not significant, $p < 0.05$ (*), $p < 0.01$ (**), and $p < 0.001$ (***).

## Supporting information

**S1 Fig. Parasite load during chronic infection, survival, and flow cytometry gating strategy in Casp1 KO mice.** (a) qPCR analysis of *T. gondii* parasite load 6 weeks post-infection (6wpi) in the brain of wildtype WT C57BL/6 (n = 9) and *Casp1* KO (n = 10) mice, two experiments. (b-c) RT-qPCR analysis of tachyzoite stage (Sag1) and bradyzoite stage (Bag1) specific genes in the brain at 6wpi, and non-stage specific gene (Act1) WT (n = 9), *Casp1* KO (n = 8). (d) Ethical end-point curve WT (n = 20), *Casp1* KO (n = 8), and *Casp1/11* KO (n = 8). (e) Weights of WT (n = 5) and *Casp1* KO (n = 5) mice throughout acute and chronic *T. gondii* infection. (f-u) Gating strategy for flow cytometry analysis of cell populations quantified in experiments. All analysis was pre-gated on singlets (f) and then on live cells (g). For T cell panels, events were gated on CD3$^+$ (h) then on either CD4$^+$ or CD8$^+$ (i) these populations were then further sub-gated on IFN-γ$^+$ (k), Tbet$^+$ (l) and CD62L$^{low}$ CD44$^{hi}$ (m). NK/ILC1 cells were gated on NK1.1$^+$CD3$^-$ and NK T cells were gated on NK1.1$^+$CD3$^+$ (j) these two sets were then gated for negative auto-fluorescent and T-bet (n and p), the final populations were gated for IFN-γ-producing cells (o and q). Myeloid cells were gated as CD11b$^+$CD45$^+$ (r) then as Ly6G+ (neutrophils) or Ly6G-negative (s), the Ly6G-negative population was then gated for Ly6C and iNOS expression (t and u). Data are presented as mean ± s.e.m., p values by randomized-block ANOVA and post-hoc Tukey test (a-c), log-rank (Mantel Cox) test (d) Two-way ANOVA and Šídák's multiple comparisons test (e).
(TIF)

**S2 Fig. Acute cytokine production and flow cytometry in Casp1 KO mice.** (a) Total spleen cell count at 8dpi, and number of CD62L$^{low}$ CD44$^{hi}$ (activated) CD4$^+$ and CD8$^+$ T cells, three experiments, WT (n = 14) and *Casp1* KO (n = 10). (b) Flow cytometry of CD3$^+$ CD8$^+$ IFN-γ$^+$ T cells in spleen at 8 dpi, three experiments, WT (n = 14) and *Casp1* KO (n = 10). (c) Frequency of splenic CD4$^+$ and CD8$^+$ T cells that are T-bet$^+$, WT (n = 14) and *Casp1* KO (n = 10). (d) Flow cytometry of CD3$^-$ NK1.1$^+$ T-bet$^+$ IFN-γ$^+$ NK cells and or ILC1 cells in spleen at 8 dpi, two experiments WT (n = 10), *Casp1* KO (n = 7). (e) Flow cytometry of CD3$^+$ NK1.1$^+$ T-bet$^+$ IFN-γ$^+$ NK T cells in spleen at 8 dpi, two experiments WT (n = 10), *Casp1* KO (n = 7). (f) Peritoneum total cell number and number of CD62L$^{low}$ CD44$^{hi}$ (activated) CD4$^+$ and CD8$^+$ T cells, three experiments, WT (n = 14), *Casp1* KO (n = 10). (g-h) Flow cytometry of CD3$^+$ (g) CD4$^+$, and (h) CD8$^+$ IFN-γ$^+$ T cells in peritoneum at 8 dpi, three experiments, WT (n = 14), *Casp1* KO (n = 10). (i) Flow cytometry of T-bet$^+$ CD4$^+$ and CD8$^+$ T cells in the peritoneum at 8 dpi, three experiments, WT (n = 14), *Casp1* KO (n = 10). (j) Flow cytometry of CD3$^-$ NK1.1$^+$ T-bet$^+$ IFN-γ$^+$ NK cells and or ILC1 cells in peritoneum at 8 dpi, two experiments WT (n = 9), *Casp1* KO (n = 6), one outlier removed from WT and one from *Casp1* KO. (k) Flow cytometry of CD3$^+$ NK1.1$^+$ T-bet$^+$ IFN-γ$^+$ NK T cells in peritoneum at 8 dpi, two experiments WT (n = 10), *Casp1* KO (n = 7). (l-n) Frequency of (l) Ly6C$^{hi}$ monocytes, two experiments, each experiments mean and SEM is plotted and genetic groups from each experiment are matched with a line, WT (n = 10), *Casp1* KO (n = 7) (m) Ly6G$^+$ neutrophils, three experiments, WT (n = 14), *Casp1* KO (n = 10) (n) iNOS$^+$ monocytes, two experiments WT (n = 10), *Casp1* KO, in peritoneum at 8 dpi. (o) Geometric mean fluorescent intensity (MFI) of iNOS expression among iNOS$^+$ monocytes, two experiments WT (n = 10), *Casp1* KO (n = 7). (p) Number of neutrophils and monocytes in peritoneum of naïve WT (n = 3) and

*Casp1* KO (n = 3) mice. (q) Frequency of neutrophils and monocytes in peritoneum of naïve WT (n = 3) and *Casp1* KO (n = 3) mice. (r-u) Frequency (r and t) and number (s and u) of CD4$^+$ and CD8$^+$ producing IFN-γ in spleens of naïve WT (n = 3) and *Casp1* KO (n = 3) mice. Data are presented as mean ± s.e.m., p values by randomized-block ANOVA and post-hoc Tukey test (a-o) or Welch's t-test (p-u).
(TIF)

**S3 Fig. Cytokine production in Il18 KO mice and IL-18R expression.** (a-b) Ex vivo cytokine release assay. (a) Percent (%) IL-18 released from equal number of PECs at 0 (n = 12), 3 (n = 4), and 6 (n = 4) dpi. (b) Total IL-18 present from equal number of PECs at 0 (n = 12), 3 (n = 4), and 6 (n = 4) dpi. (c) Number of cells in the spleen 8 dpi. (d) Flow cytometry of CD3$^+$ CD8$^+$ IFN-γ$^+$ T cells in spleens 8 dpi. (e) Number of cells in the peritoneum 8 dpi. (f,g) Flow cytometry analysis of frequency (f) and number (g) of CD3$^+$ CD4$^+$ IFN-γ$^+$ T cells in perito-neum 8 dpi. (h) Flow cytometry number of CD3$^+$ CD8$^+$ IFN-γ$^+$ T cells in peritoneum 8 dpi. (i-k) Frequency of (i) Ly6C$^{hi}$ monocytes, (j) Ly6G$^+$ neutrophils, and (k) iNOS$^+$ monocytes in peritoneum at 8 dpi. (l) MFI of iNOS expression among iNOS$^+$ monocytes. (c-l) three experi-ments WT (*n* = 15) and *Il18* KO (*n* = 13). (m) Flow-gating strategy for IL-18 receptor (IL-18R) expression in CD4$^+$ and CD8$^+$ T cells. (n) Flow cytometry analysis of splenic CD3$^+$ CD4$^+$ and CD8$^+$ T cells IL-18R expression, two experiments, (*n* = 20). (o) Flow cytometry analysis of peritoneal CD3$^+$ CD4$^+$ and CD8$^+$ T cell IL-18R expression, one experiment, (*n* = 10). (p) Flow cytometry analysis of splenic CD3$^+$ CD4$^+$ and CD8$^+$ T cells, NK cells and or ILC1 cells (CD3-NK1.1+ Tbet+) and NK T cells (CD3+ NK1.1+ Tbet+) IL-18R expression in naïve (*n* = 5) and infected mice (*n* = 5). Data are presented as mean ± s.e.m., p values by one-way ANOVA and post-hoc Tukey test (a and b), randomized-block ANOVA and post-hoc Tukey test (c-l and n) or Student's t-test (o-p).
(TIF)

**S4 Fig. Expression of cell death-related genes using single-cell RNAseq.** (a-b) Analysis of single cell RNA sequencing of the spleen on day 14 post-infection from [30] (a) Cell cluster segmentation markers (b) Cell type expression of manually selected inflammasome and cyto-kine related genes.
(TIF)

**S5 Fig. Parasite burden and cytokine production in Cx3cr1$^+$ Casp1 KO mice.** (a) Cyst counts of *T. gondii* parasite load 6 weeks post-infection (6wpi) in the brain of wildtype WT (n = 9) and *Cx3cr1$^+$ Casp1* KO (n = 11) mice, two experiments. (b) Parasite load in the spleen 8dpi in WT (n = 4) and *Cx3cr1$^+$ Casp1* KO (n = 4) mice. (c and d) Ex vivo cytokine release assay WT (n = 3) and *Cx3cr1$^+$ Casp1* KO (n = 4). (c) Total IL-18 present from equal number of PECs at 6 dpi. (d) Percent (%) IL-18 released from equal number of PECs at 6 dpi. (e) Splenic cell number and number of CD62L$^{low}$ CD44$^{hi}$ (activated) CD4$^+$ and CD8$^+$ T cells 8dpi, two experiments, WT (n = 8) and *Cx3cr1$^+$ Casp1* KO (n = 9). (f) Flow cytometry of CD3$^+$ CD8$^+$ IFN-γ$^+$ T cells in spleen at 8 dpi, two experiments, WT (n = 8) and *Cx3cr1$^+$ Casp1* KO (n = 9). (g) T-bet$^+$ splenic CD4$^+$ and CD8$^+$ T cells 8dpi, two experiments, WT (n = 8) and *Cx3cr1$^+$ Casp1* KO (n = 9). (h) Frequency IFN-γ$^+$ splenic CD4$^+$ and CD8$^+$ T cell in naïve WT (n = 3) and *Cx3cr1$^+$ Casp1* KO (n = 3) mice. (i) Peritoneal exudate cell number and number of CD62L$^{low}$ CD44$^{hi}$ (activated) CD4$^+$ and CD8$^+$ T cells 8dpi, two experiments, WT (n = 10) and *Cx3cr1$^+$ Casp1* KO (n = 7). (j-k) Flow cytometry of CD3$^+$ (j) CD4$^+$ (k) CD8$^+$ IFN-γ$^+$ T cells in peritoneum at 8 dpi, two experiments, WT (n = 10) and *Cx3cr1$^+$ Casp1* KO (n = 7). (l) T-bet$^+$ peritoneal CD4$^+$ and CD8$^+$ T cells 8dpi, two experiments, WT (n = 10) and *Cx3cr1$^+$ Casp1* KO (n = 7). (m-n) Frequency of iNOS$^+$ monocytes in peritoneum(m) and Geometric mean

intensity of iNOS among iNOS+ monocytes(n). Data are presented as mean ± s.e.m., p values by randomized-block ANOVA and post-hoc Tukey test (a, e-g, i-n) and Welch's t-test (b, c, d and h).
(TIF)

**S6 Fig. Characterization of Cx3cr1+ cell populations.** (a-h) Inducible Cx3cr1CreERT2/WT Rosa26 Ai6/Ai6 (Cx3cr1-reporter mice) were given tamoxifen chow and injected with PBS or 10 cysts Me49. (a-b) Reprehensive flow plots of Cx3cr1+ cells (ZsGreen-positive) (a) Naïve reporter mouse PECs (b) Reporter mouse PECs 6dpi. (c) ZsGreen+ cells from naïve and 6dpi were gated on CD45+ CD11b-negative or CD11b-positive. (d-e) CD45+CD11b+ cells were gated on F480 and Ly6C expression. (f) Quantification of % of ZsGreen (Cx3cr1) positive cells with/without F480 expression and with or without Ly6C expression, naïve (n = 4), 3dpi (n = 3), and 6dpi (n = 3). (g) Quantification of % of ZsGreen (Cx3cr1) positive cells CD3+ or Ly6G+, naïve (n = 4), 3dpi (n = 3), and 6dpi (n = 3). (h) qPCR heatmap for Casp1, Il18, Gbp2, Ifngr1 in Cx3cr1+ F480(+/-) and Ly6C(+/-) myeloid populations at 0, 3, and 6 dpi. Shown as $\log_2$ change relative to 0 dpi F480+ Ly6C-. Data are presented as mean ± s.e.m., p values by Welch's t-test (f and g).
(TIF)

**S7 Fig. IL-18 release and downstream responses are not dependent on Stat1 signaling.** (a-g) Cx3cr1 CreERT2/WT Stat1WT/WT and Cx3cr1 CreERT2/WT Stat1fl/fl mice were administered 5 doses of tamoxifen i.p. at 20 g/kg. After two weeks, 10 cysts of Me49 was administered i.p. (a) Peritoneal parasite burden at 8 dpi. (b) IL-18 and IFN-γ cytokine levels in serum and peritoneal fluid (Per-Fluid) at 8 dpi. (c) qPCR of Casp1 and Il18 at 8 dpi in PECs. (d-g) Flow cytometry of splenocytes incubated in BFA for 5 hours. (d) Percent of CD4+ T cells making IFN-γ. (e) Number of CD4+ T cells making IFN-γ. (f) Percent of CD8+ T cells making IFN-γ. (g) Number of CD8+ T cells making IFN-γ. Data are presented as mean ± s.e.m., p values by Welch's t-test (a-g).
(TIF)

**S1 Data. Supporting file.** Raw data for S1e weight loss and S6h qPCR.
(XLSX)

## Author Contributions

**Conceptualization:** Isaac W. Babcock, Michael A. Kovacs, Tajie H. Harris.

**Data curation:** Isaac W. Babcock, Ish Sethi.

**Formal analysis:** Isaac W. Babcock, Maureen N. Cowan, Seblework Alemu.

**Funding acquisition:** Tajie H. Harris.

**Investigation:** Isaac W. Babcock, Lydia A. Sibley, Ish Sethi, Seblework Alemu.

**Methodology:** Isaac W. Babcock, Maureen N. Cowan, Seblework Alemu.

**Project administration:** Tajie H. Harris.

**Resources:** John R. Lukens, Tajie H. Harris.

**Supervision:** Tajie H. Harris.

**Validation:** Isaac W. Babcock.

**Visualization:** Isaac W. Babcock, Maureen N. Cowan.

**Writing – original draft:** Isaac W. Babcock.

**Writing – review & editing:** Isaac W. Babcock, Lydia A. Sibley, Sydney A. Labuzan, Abigail G. Kelly, John R. Lukens, Tajie H. Harris.

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
