## [Decision Letter · Decision Letter 0]

4 Mar 2024

Dear Dr. Harris,

Thank you very much for submitting your manuscript "Caspase-1 in Cx3cr1-expressing cells drives an IL-18-dependent T cell response that promotes parasite control during acute T. gondii infection" for consideration at PLOS Pathogens. As with all papers reviewed by the journal, your manuscript was reviewed by members of the editorial board and by several independent reviewers. In light of the reviews (below this email), we would like to invite the resubmission of a significantly-revised version that takes into account the reviewers' comments.

Several important issues were raised by each reviewer. These will need to be addressed by inclusion of additional experimental data.

We cannot make any decision about publication until we have seen the revised manuscript and your response to the reviewers' comments. Your revised manuscript is also likely to be sent to reviewers for further evaluation.

Sincerely,

Eric Y Denkers

Academic Editor

PLOS Pathogens

James Collins III

Section Editor

PLOS Pathogens

Michael Malim

Editor-in-Chief

PLOS Pathogens

orcid.org/0000-0002-7699-2064

Several important issues were raised by each reviewer. These will need to be addressed by inclusion of additional experimental data.

Reviewer's Responses to Questions

**Part I - Summary**

Reviewer #1: s is an interesting study where the authors report that the critical function of Caspase 1 in CX3CR1 expressing macrophages is to drive IL18 release to maintain CD4+ T cell IFN-γ production and play a protective role during acute T. gondii infection. Previously the same group reported that IL-1α release from microglia (CX3CR1 positive cells) promotes myeloid cells recruitment into the infected brain and controls parasites during chronic T. gondii infection (Batista et al, 2020, Nat Communications). However, in the current study they did not observe myeloid cell recruitment. Overall, the manuscript is well written, and the conclusions are supported by the data. There are some concerns that need to be addressed by the authors as mentioned below.

Reviewer #2: The study performed by Babcock et al. focuses on examining how a deficiency in Caspase-1 may affect the immunity in Toxoplasma gondii infected mice. The authors observed only an increase in the parasite burden that was, nevertheless, well tolerated since survival was unimpaired and infection progressed to the chronic phase.

The authors did not test if a higher dose of the parasite and/or oral route of infection will have a more significant impact on host protective immune responses in infected Caspase-1 deficient animals.

The data presented have several gaps. While the parasite burden is higher in the peritoneum (site of infection), the authors focused on a small change in IFN-gamma producing CD4+ T lymphocyte in the spleen. It appears that IFN-g+ CD4+ were analyzed without any restimulation, but also without prior in vivo Brefeldin A injection, which might not be the most accurate way to examine their effector function. According to Material and Methods, the authors used anti-Tbet Ab in their panel, as well as CD44 and CD62L. It would be important to show whether there is a difference in the number of activated CD44+ CD4+ T cells. What are the frequencies of T-bet+ populations?

Similarly, while the frequency of IFN-g+ CD4+ T cells appears to be only slightly decreased, their absolute number is much more affected. Is spleen smaller in Caspase-1 KO mice when compared to ones from WT group on day 8 post-infection? What was the parasite burden in spleens from WT and KO animals?

Next, authors show that in the peritoneum there is a decreased number of monocytes and neutrophils. Again, although they employed anti-iNOS Ab in their panel (line 457), we don’t know if the production of iNOS was diminished, which could contribute to increased parasite load in Caspase-1 deficient group. What was the frequency of the recruited inflammatory macrophages in peritoneum? A more comprehensive phenotyping of cells in peritoneum as indicated above for splenocytes needs to be provided. Interestingly, the ratio between number of monocytes vs. neutrophils looks quite comparable between the WT and Caspase-1 KO animals (~25:1).

Sometimes minor changes that can reach statistical significance may not have biological significance. A similar drop in the frequency of neutrophils was observed in infected IL-1RAP KO as in Caspase-1 KO mice but with no impact on the parasite load. Moreover, the “lower” number of neutrophils in infected IL-1RAP KO mice (Fig. 3) is the same as in infected control WT group in Figure 1.

In the rest of the manuscript, authors explore the role of IL-18 during T. gondii infection. They show that the mice lacking IL-8, as in Caspase-1 KO group, have decreased (-20%) serum IFN-gamma levels, which correlates with a smaller pool of IFN-gamma+ CD4+ T cells in spleen. The same observation (Fig. 2 vs 4) was made in mice that lack expression of Caspase-1 in Cx3cr1+ cells targeted as a potential source of IL-18. However, in the latter group, substantial systemic levels of IL-18 remain detectable (in contrast to IL-18 KO animals; much higher than in Caspase-1 KO and not very different from the level observed in WT control group in Fig. 1d).

Moreover, exogenous IL-18 administration augmented neither serum IL-18 nor IFN-gamma levels in mice that lack expression of Caspase-1 in Cx3cr1+ cells, but it did increase the frequency of IFN-gamma+ CD4+ T cells in spleen that correlates with decreased parasite burden in peritoneum.

Based on the well-documented IFN-gamma-stimulatory role of IL-18, it is expected that rIL-18 would improve the Th1 immunity in mice that lack expression of Caspase-1 in Cx3cr1+ cells; very likely rIL-12 treatment would do the same. It is not clear why the authors didn’t try to rescue IL-18 deficient animals with rIL-18 treatment.

In addition, because IFN-gamma regulates the transition of inflammatory monocytes to Cx3cr1+ macrophages, it is important to show the frequency of the latter population throughout the study. Otherwise, one can argue that, due to some other yet-to-be defined defect, the infected Crisper-1 KO mice are making less IFN-gamma that impairs development of Cx3cr1+ macrophages, which leads to lower serum levels of IL-18.

In Figures 4 and 5, mice that lack expression of Caspase-1 in Cx3cr1+ population are referred to as Mf Casp KO (for simplicity?). Is it or is it not Mf Casp KO? After reading the manuscript, the tittle “Caspase-1 producing Cx3cr1+ cells…” sounds ambiguous. The authors should stick to one or the other interpretation. Once more, better quantification of lymphoid and myeloid populations in spleen and peritoneum from mice that lack Caspase-1 expression in Cx3cr1+ cells would strengthen this manuscript. For example: How many Cx3cr1+ myeloid cells are in spleen vs. peritoneum and in KOs vs. WT mice?

Reviewer #3: Here, Babcock, Harris and colleagues investigate the role of caspase-1 in the immune response to live T. gondii infection. Using both whole body and cell-type restricted deficiency models, they conclude that caspase-1 is required for optimal immune responses and the effect of caspase-1 is principally mediated by IL-18 produced and released from a CX3CR1-expressing cell.

The chief consequence observed downstream of caspase-1 deficiency seems relatively subtle/minor in nature, ie., subtle increases in parasite loads in the acute and chronic phases of infection. It is not clear whether the increase is parasite loads is mainly an increase in bradyzoite loads or also accompanied by increased tachyzoite numbers in the CNS. Given that no changes were observed in overall mortality, it is more likely that the mice were able to achieve sufficient control of chronic infection, without a clear increase in reactivation of lytic phase infection.

**Part II – Major Issues: Key Experiments Required for Acceptance**

Reviewer #1: 1. To confirm that in naïve mice lacking Caspase 1, IL18, cx3cr1 expressing macrophages had no effect on the presence of CD4+ T cell IFN-γ in the spleen it is strongly recommended to add flow cytometry results before infection (Day 0) in Fig 1e, 2d and 4e.

2. To confirm that Casp1 in CX3CR1 expressing macrophages play a major role in the regulation of CD4+ T cell IFN-γ the authors should perform an adoptive transfer of CX3CR1 expressing macrophages into Casp1 KO mice to check if it restores CD4+ T cell IFN-γ production and control parasite burden.

3. To know during in vivo T. gondii infection if direct invasion of the parasites is required or host sensing/ parasite attachment/protein secretion is sufficient for Caspase 1 activation to release IL-18 the authors should perform a similar experiment as Fig 1f using heat killed parasites.

4. Authors should include a gating strategy which was used to define different immune cells populations (T cells, Mono, Neutrophils, NK/ ILC1/ NK T cells) in the supplementary figure for better understanding of the reader.

Clarifications not requiring data

5. it seems possible that the microbiota has a role in the secretion of IL-18 by caspase 1. This could be addressed by examining the parasite burden in wild type C57Bl6 and caspase 1 ko mice with and without antibiotics. Alternatively, the authors should acknowledge this limitation and suggest follow up studies in the Discussion.

6. Please explain in the discussion why CD4+ T cell IFN-γ production is significant in Casp1 KO mice in spleen but not in the peritoneum (Fig 1 and Suppl Fig S1).

Reviewer #2: 1. A detailed FACS analyses of lymphoid and myeloid populations in peritoneum and spleen of infected the mice is required.

2. Characterize the population of Caspase-1 expressing Cx3cr1+ cells.

3. Demonstrate that IL-18 expression is upstream (of IFN-g) and not down-stream from IFNgR-gamma signaling.

Reviewer #3: The differences in IFN-g levels were reported to be selectively significant for CD4 T cells (Figure 1F), but not for other cell types (supplementary data). However, a closer inspection of the data shows that there may also be changes (increases/decreases) in CD8 T cells, NK and ILCs which may not have reach statistical significance. It would be better to display the p-values, rather than denoting these as non-significant. The overall decreases in IFN-g levels is about 20-30%, but the levels produced in caspase-deficient mice are still very high and apparently sufficient to control acute infection and allow for establishment and maintenance of chronic infection. Thus, it is not clear to what extent the additional production of IFN-g by CD4 T cells is beneficial overall to host resistance/disease tolerance. The flip side of the argument is that increased IFN-g production by CD4 T cells induced by caspase-1/IL-18 likely increases acute immunopathology and could further contribute to inflammation of the CNS in the chronic phase. Did the authors track the weight changes during infection and did the authors examine whether caspase-1/IL-18 deficiency ameliorated chronic symptoms?

The authors utilized the Cx3cr1 cre system to delete Caspase 1 and call them “macrophage Casp1 knockout” mice. However, they do not clearly delineate the cell types affected by this maneuver nor discuss how to interpret the results obtained. Which specific myeloid/lymphoid cell derived lineages are affected here? cDC-1, CCR2+ monocytes and their derivatives and tissue macrophages?

**Part III – Minor Issues: Editorial and Data Presentation Modifications**

Reviewer #1: 1. Title- The main finding of the study is the role of Caspase in Cx3cr1 expressing macrophages during infection so, it would be better to have cx3cr1- expressing macrophages in the title. Also change T. gondii to Toxoplasma gondii.

2. Line 146- Correct ICL1 to ILC1.

3. Line 164- Add a short background to explain the rationale of the experiment. “A recent study from Clark et. al., 2023, Cell Report describes that IL-18, and its regulation impacts the ability of CD4+ T cells to make IFN-γ”.

4. Line 234- Change S3b-S3d Fig to S3d-S3f.

5. Line 270- A study by Sateriale et al, 2020, PNAS has shown that caspase-1 is sufficient to provide innate resistance to cryptosporidiosis when compared to Casp1/11 KO mice. This reference can be added in the discussion to support your study.

6. In the Figure 3i- change heading from Ly6G+ to Neutrophils.

7. Line 704- In the figure legend change (j) to (g).

Reviewer #2: Figure formatting should be improved (e.g. Figure 2: Peritoneum day 8 post ?).

Using scientific notation would also be helpful when showing absolute cell numbers.

Reviewer #3: (No Response)

PLOS authors have the option to publish the peer review history of their article (what does this mean?). If published, this will include your full peer review and any attached files.

Reviewer #1: No

Reviewer #2: No

Reviewer #3: No
---

## [Decision Letter · Decision Letter 1]

27 Jul 2024

Dear Dr. Harris,

Thank you very much for submitting your manuscript "Caspase-1 in Cx3cr1-expressing cells drives an IL-18-dependent T cell response that promotes parasite control during acute Toxoplasma gondii infection" for consideration at PLOS Pathogens. As with all papers reviewed by the journal, your manuscript was reviewed by members of the editorial board and by several independent reviewers. In light of the reviews (below this email), we would like to invite the resubmission of a significantly-revised version that takes into account the reviewers' comments.

Two of the reviewers are enthusiastic about this manuscript, but the remaining reviewer raises some valid concerns. In particular, the two issues of the nature of the Caspase 1 CXCR3+ cells and whether IL-18 is up or downstream of IFN-g are both important to address.

We cannot make any decision about publication until we have seen the revised manuscript and your response to the reviewers' comments. Your revised manuscript is also likely to be sent to reviewers for further evaluation.

Sincerely,

Eric Y Denkers

Academic Editor

PLOS Pathogens

James Collins III

Section Editor

PLOS Pathogens

Michael Malim

Editor-in-Chief

PLOS Pathogens

orcid.org/0000-0002-7699-2064

Two of the reviewers are enthusiastic about this manuscript, but the remaining reviewer raises some valid concerns. In particular, the two issues of the nature of the Caspase 1 CXCR3+ cells and whether IL-18 is up or downstream of IFN-g are both important to address.

Reviewer's Responses to Questions

**Part I - Summary**

Reviewer #2: TThe revisions of the manuscript by Babcock et al. only marginally address the main questions/concerns raised by the reviewers. Although the editors asked for Major Revisions, none of the 5 main Figures has been modified and/or improved with new information. New information provided in Supplementary Figures hardly enhances the original data and raises even more questions.

The nature of the Cx3cr1 population remains obscure. To make things even more complicated, the authors add on a possibility that steady-state expressing Cx3cr1 cells are responsible for IL-18 production (although in the experiment shown in Response Figure 1, the frequency of IFN-g+ CD4 T lymphocytes was not significantly reduced in spleen, in contrast to observations made in infected Casp1 KO mice). Moreover, based on data in Figure S5, Cx3cr1 expressing cells in PEC and spleen in infected mice are very different from the ones in steady-state (as expected inflammatory monocytes vs tissue resident macrophages). Again, the Casp1 expressing population could be the former or latter, both, or neither.

Similarly, based on Figure S2, the defect in Casp1 KO mice could be due to the lack of splenic CD4+ Th1 lymphocytes, or to an increase in frequency of CD8+ T cells that express Tbet (observed both in spleen and PEC). We just do not know.

Differences in responses between PEC and spleen specifically in Casp1 KO or Cx3cr1+ Casp1 KO (but not IL-18 KO) infected with T. gondii remain poorly explained. Perhaps instead of looking only at one time point, the authors should have performed a kinetics study with earlier time points (e.g. day 3, 5 and 8) and examined all possible sources of IFN-g (e.g. ILC, gd T, NK, cells).

I agree with the authors that there is too much variation between individual data points, which may potentiate minor and/or cancel out real differences between tested groups when results from several experiments are pooled.

Reviewer #3: The authors have responded appropriately to the previous comments. I do not have any further comments.

Reviewer #4: This is a very thorough revised manuscript that provides important information regarding the role of CX3CR1-positive macrophages in regulating caspase-1 dependent IL-18 response during T. gondii infection. Not only did the authors develop precise mouse models to clarify a very confusing field, but they also defined the role of caspase-1 during T. gondii infection. These are undoubtedly important and impactful results. In addition, while this reviewer did not evaluate the initial submission, I feel that the authors fully addressed the previously identified deficiencies.

**Part II – Major Issues: Key Experiments Required for Acceptance**

Reviewer #2: The authors did not satisfactory addressed two major issues:

2. Characterize the population of Caspase-1 expressing Cx3cr1+ cells.

3. Demonstrate that IL-18 expression is upstream (of IFN-g) and not down-stream from IFNgR-gamma signaling.

Reviewer #3: (No Response)

Reviewer #4: (No Response)

**Part III – Minor Issues: Editorial and Data Presentation Modifications**

Reviewer #2: If n=1, it cannot be a representative graph.

Figure S5 (j) Representative graph of splenic ZsGreen (Cx3cr1) positive cells from naïve (n=1) and 8dpi (n=2) showing their distribution within varying F480 and Ly6C expressing populations.

Panel K in Figure S5 might be out of place (?). This panel, in agreement with data depicted in the Figure 1 panel h, shows a lower frequency of inflammatory monocytes, the essential effectors population for control of T. gondii replication. However, this was not observed in infected Cx3cr1 Cas1 KO mice (figure 4h), again showing that correlations between different mouse strains and sets of pre-selected immunological parameters cannot be used as evidence for causation.

The conclusion that Casp1 KO mice do not lose weight during the infection is based on 1 time point (day 35 post infection).

Using scientific numbers for the graphs that depicts cell numbers in millions should be applied in all such panels.

Reviewer #3: (No Response)

Reviewer #4: I have only two minor recommendations for the authors:

1.Line 86: Please add ILC1 as IFN-γ producers downstream of IL-12 and provide the appropriate references for T. gondii infections.

2.One puzzle (no need to address experimentally) is the relatively large amount of IL-18 in CX3CR1xCasp1fl/fl mice (Fig. 4D). This suggests that while CX3CR1-positive macrophages are contributing to IL-18 production, other cells can produce this cytokine as well. There is no need to perform any additional experiments; just acknowledge this fact and compare the data to the previously published Casp1x11 complete KO mice.

PLOS authors have the option to publish the peer review history of their article (what does this mean?). If published, this will include your full peer review and any attached files.

Reviewer #2: No

Reviewer #3: No

Reviewer #4: **Yes: **Felix Yarovinsky
---

## [Editor Report · Decision Letter 2]

13 Oct 2024

Dear Dr. Harris,

We are pleased to inform you that your manuscript 'Caspase-1 in Cx3cr1-expressing cells drives an IL-18-dependent T cell response that promotes parasite control during acute Toxoplasma gondii infection' has been provisionally accepted for publication in PLOS Pathogens.

Best regards,

Eric Y Denkers

Academic Editor

PLOS Pathogens

James Collins III

Section Editor

PLOS Pathogens

Michael Malim

Editor-in-Chief

PLOS Pathogens

orcid.org/0000-0002-7699-2064

The authors have now examined the nature of Caspase 1+ CXCR3+ cells through cell sorting followed by PCR transcript analysis. They have performed new experiments providing evidence that IL-18 lies upstream of IFN-gamma signaling. Therefore, the two concerns raised by the reviewer have been addressed.
---

## [Editor Report · Acceptance letter]

20 Oct 2024

Dear Dr. Harris,

We are delighted to inform you that your manuscript, "Caspase-1 in Cx3cr1-expressing cells drives an IL-18-dependent T cell response that promotes parasite control during acute *Toxoplasma gondii* infection," has been formally accepted for publication in PLOS Pathogens.

Best regards,

Michael Malim

Editor-in-Chief

PLOS Pathogens

orcid.org/0000-0002-7699-2064